# X-NeMo: Expressive Neural Motion Reenactment via Disentangled Latent Attention

**Xiaochen Zhao**[1,2]*, **Hongyi Xu**[2] , **Guoxian Song**[2], **You Xie**[2], **Chenxu Zhang**[2], **Xiu Li**[2],
**Linjie Luo**[2], **Jinli Suo**[1], **Yebin Liu**[1]†
[1] Tsinghua University, [2] ByteDance Inc.

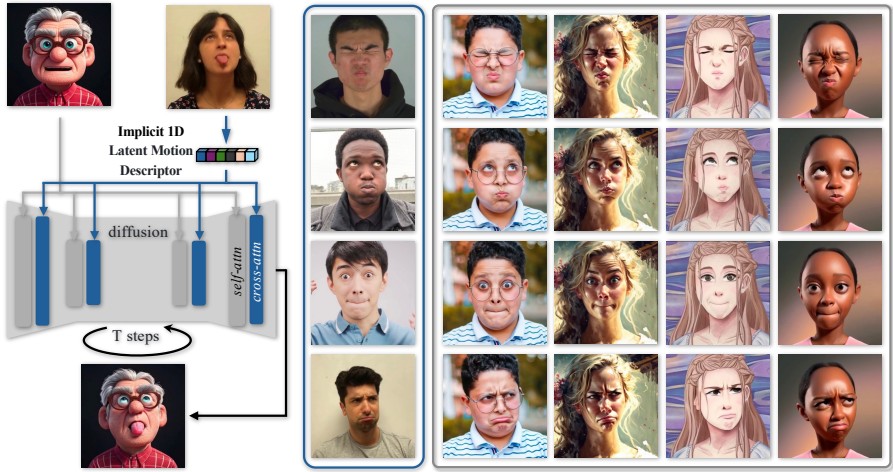

Figure 1: We present X-NeMo, a diffusion-based portrait animation framework that integrates expressive 1D latent motion descriptors with identity-disentangled motion control through cross-attention mechanisms (left). Our method enables meticulous transfer of expressive head poses and detailed facial expressions while maintaining identity consistency, even across subjects with distinct appearances, styles and facial structures (right).

## Abstract

We propose X-NeMo, a novel zero-shot diffusion-based portrait animation pipeline that animates a static portrait using facial movements from a driving video of a different individual. Our work first identifies the root causes of the key issues in prior approaches, such as identity leakage and difficulty in capturing subtle and extreme expressions. To address these challenges, we introduce a self-supervised training framework that distills a 1D identity-agnostic latent motion descriptor from driving image, effectively controlling motion through cross-attention during image generation. Our implicit motion descriptor captures expressive facial motion in fine detail, learned from a diverse video dataset without reliance on pretrained motion detectors. We further enhance expressiveness and disentangle motion latents from identity cues by supervising their learning with a dual GAN decoder, alongside spatial and color augmentations. By embedding the driving motion into a 1D latent vector and controlling motion via cross-attention rather than additive spatial guidance, our design eliminates the transmission of spatial-aligned structural clues from the driving condition to the diffusion backbone, substantially mitigating identity leakage. Extensive experiments demonstrate that X-NeMo surpasses state-of-the-art baselines, producing highly expressive animations with superior identity resemblance. Our code and models will be available for research at our project page.

---

*Work done during the internship at ByteDance.
†Corresponding author.

# 1 INTRODUCTION

We investigate the task of portrait animation, where a static portrait is animated using head movements and facial expressions derived from a driving video of a different subject. This task has garnered growing interest owing to its versatile applications in video conferencing, visual effects and digital agents. Building on prior research, we aim to advent the field of zero-shot portrait reenactment by synthesizing *highly expressive* animations while maintaining *identity resemblance* to the reference portraits with minimal loss.

Commencing with the pioneering works Siarohin et al. (2019b;a), portrait animation has primarily involved extracting motion features from a driving video followed with a generative process, such as GANs Goodfellow et al. (2014); Karras et al. (2019; 2020) or diffusion models Ho et al. (2020); Song et al. (2020a;b); Rombach et al. (2022), conditioned on the reference appearance and derived motion features. Recent advancements in diffusion models have achieved unprecedented diversity and quality in image generation, prompting us to utilize their generative capabilities Saharia et al. (2022); AI (2022) for portrait animation. Recent approaches have tackled portrait animation as a controlled image-to-video diffusion task, where the reference appearance is cross-queried through mutual self-attention Cao et al. (2023) whereas the driving motion signal is integrated into the denoising process using frameworks like ControlNet Zhang et al. (2023b) or lighter-weight PoseGuider Hu et al. (2023). The driving motion is represented either through explicit semantic signals such as facial landmarks Ma et al. (2024); Wei et al. (2024); Chang et al. (2024), dense pose Xu et al. (2024d) and facial template renderings Chen et al. (2024a), or through implicit motion features learnt from synthetic cross-identity image pairs with aligned expression but different identities Xie et al. (2024); Yang et al. (2024a). Despite significant progress in realism and dynamics, these diffusion-based methods still struggle to capture extreme or subtle expressions and often suffer from identity drifting, particularly when the reference and driving identities differ substantially.

We identify two main factors contributing to the challenges in expressiveness and identity resemblance in prior network designs. First, explicit motion descriptors like facial landmarks or blendshapes, are often too coarse to capture extreme or subtle facial motions and rely heavily on the robustness and accuracy of external motion detectors. Although these descriptors do not contain RGB appearance, they encode the facial structure of the driving identity, leading to undesirable identity leakage in cross-identity animations. Recent approaches Xie et al. (2024); Yang et al. (2024a) have attempted to derive implicit motion signals directly from synthetic cross-identity training image pairs generated using an off-the-shelf portrait animator (e.g., Wang et al. (2021)). Despite substantial improvements in expressiveness and stability, these methods remain constrained by the capacity of pretrained portrait animators which struggle with complex expressions (e.g., tongue protrusion, cheek puffing). Additionally, sharing aligned facial structures in training pairs inadvertently pass identity information onto the learnt implicit motion features. Second, prior diffusion-based approaches often guide motion control using spatially-aligned 2D conditions via ControlNet or PoseGuider. While effective for self-driven motion, this approach encourages the diffusion backbone to take a shortcut to mimic the 2D layout rather than fully leveraging semantic mappings between reference and driving images, leading to identity leakage during expression transfer across different subjects.

In this work, we propose X-NeMo, a novel portrait animation framework that enables self-supervised learning of a compact 1D latent motion descriptor, facilitating effective motion control in diffusion models via *cross attentions*. Specifically, we introduce a motion encoder to extract a 1-D *identity-agnostic* motion latent from the original driving image, and modulate this controlling motion descriptor into the diffusion backbone via cross-attentions. By training jointly with the image generator, our encoder fully leverages the motion diversity and richness embedded in our training video collections, without reliance on off-the-shelf motion detectors. We restrict the dimensionality of the latent embedding, functioning as a low-pass filter Burkov et al. (2020), and format it as a 1D global motion descriptor that excludes 2D structural cues from the driving image. Furthermore, by using cross motion attentions rather than spatial additive guidance, we ensure that the backbone remains agnostic to the identity structural signals from the motion control branch. This structure-agnostic motion control enables various augmentations like color jittering and spatial transformations, promoting the self-supervised disentanglement of identity and motion. In addition to the diffusion loss, we incorporate a dual GAN-based decoder head and refine the learning of our motion latent space with image-level losses that capture subtle and detailed expressions. Our design effectively mitigates the aforementioned shortcut learning, and compels the network to interpret fine-grained motion semantics during both motion encoding and image generation stages.

Trained on a collective of public video datasets Zhang et al. (2021); Xie et al. (2022); Kirschstein et al. (2023), our method excels at faithfully capturing both extreme and nuanced facial motions and transferring them across subjects even with distinct identity attributes. We extensively evaluate our model across our challenging benchmarks and X-NeMo outperforms state-of-the-art portrait animation baselines both quantitatively and qualitatively. Additionally, our expressive latent motion descriptor serves as a unified identity-agnostic embedding, facilitating motion interpolation and video outpainting applications beyond portrait animation. We summarize our contributions as follows,

- A novel diffusion-based portrait animation pipeline, coupled with latent motion representation, achieving state-of-the-art performance in terms of motion accuracy and identity disentanglement.
- A structure-agnostic motion control scheme that learns a 1-D identity-disentangled latent motion descriptor and modulates control into image generation via cross-attentions, effectively addressing the long-standing issues of identity entanglement and motion expressiveness loss.
- A set of carefully designed strategies during both training (e.g., dual head latent supervision, augmentations and reference feature masking) and inference that substantially enhance the model performance, as supported by extensive ablation studies.
- Demonstration of captivating zero-shot portrait animations and generations.

## 2 RELATED WORKS

**GAN-based Portrait Animation.** Video-driven face reenactment seeks to accurately transfer facial expressions and head movements from a driving video to a target image. Common approaches have primarily leveraged Generative Adversarial Networks (GANs) Goodfellow et al. (2014); Karras et al. (2019; 2020) to model and capture the intricate motion dynamics between source and target identities. Broadly, these methods can be categorized into two classes: The first class is based on *explicit motion representations* Siarohin et al. (2019a;b); Ren et al. (2021); Wang et al. (2021); Mallya et al. (2022); Yin et al. (2022); Gao et al. (2023); Doukas et al. (2021); Guo et al. (2024a); Zhao & Zhang (2022), such as 3D face model parameters, landmarks, or latent keypoints, which use structured information to disentangle appearance and motion, but struggle with large pose changes or dynamic expressions. The second category involves *latent motion representations* Burkov et al. (2020); Liang et al. (2022); Zhou et al. (2021); Pang et al. (2023); Wang et al. (2022; 2023); Drobyshev et al. (2022; 2024); Xu et al. (2024c), embedding motion information in a latent space, offering improved expressiveness but relying on complex loss functions and hyperparameters to achieve identity-motion disentanglement. While more effective at transferring subtle expressions, such methods are still limited by the capability of GAN-based generators in handling extreme expressions and out-of-domain portrait styles. Our work follows this disentangled representation learning approach, but instead we use a Diffusion Model as the generator, which offers significantly improved generation capabilities with diverse and complex portrait styles.

**Diffusion-based Portrait Animation.** Diffusion models Ho et al. (2020); Song et al. (2020a;b) have demonstrated strong generative capabilities, with Latent Diffusion Models (LDMs) Rombach et al. (2022) further advancing its efficiency by operating in a lower-dimensional latent space. Recent works Liu et al. (2024); Xu et al. (2024a); Han et al. (2023); Varanka et al. (2024); Paskaleva et al. (2024) have explored adapting pre-trained LDMs AI (2022) for conditional portrait generation, by mapping reference images and driving signals into the text embeddings (e.g., using CLIP Khandelwal et al. (2022)) and injecting them into cross-attention layers. While effective for coarse-level facial expression editing, these methods still struggle with appearance and motion consistency in portrait video animation. Hu et al. (2023); Xu et al. (2024d); Chang et al. (2024) designed for human body animation have shown that the combination of a dual U-Net with mutual self-attention Cao et al. (2023) and temporal module Guo et al. (2024b) is able to maintain motion smoothness with consistent appearance. This framework has been extended to portrait animation in several works Tian et al. (2024); Xie et al. (2024); Wei et al. (2024); Xu et al. (2024b); Yang et al. (2024a); Wang et al. (2024); Ma et al. (2024); Chen et al. (2024b), often using ControlNet Zhang et al. (2023b) or PoseGuider Hu et al. (2023) for motion control. During training, they rely on explicit representations like facial keypoints Ma et al. (2024); Wei et al. (2024), facial mesh renderings Chen et al. (2024a), or synthetic cross-identity portraits Xie et al. (2024); Yang et al. (2024a). In contrast, our method learns a latent motion representation jointly with our diffusion backbone, and incorporates motion control with cross-attentions, effectively preventing identity leakage.

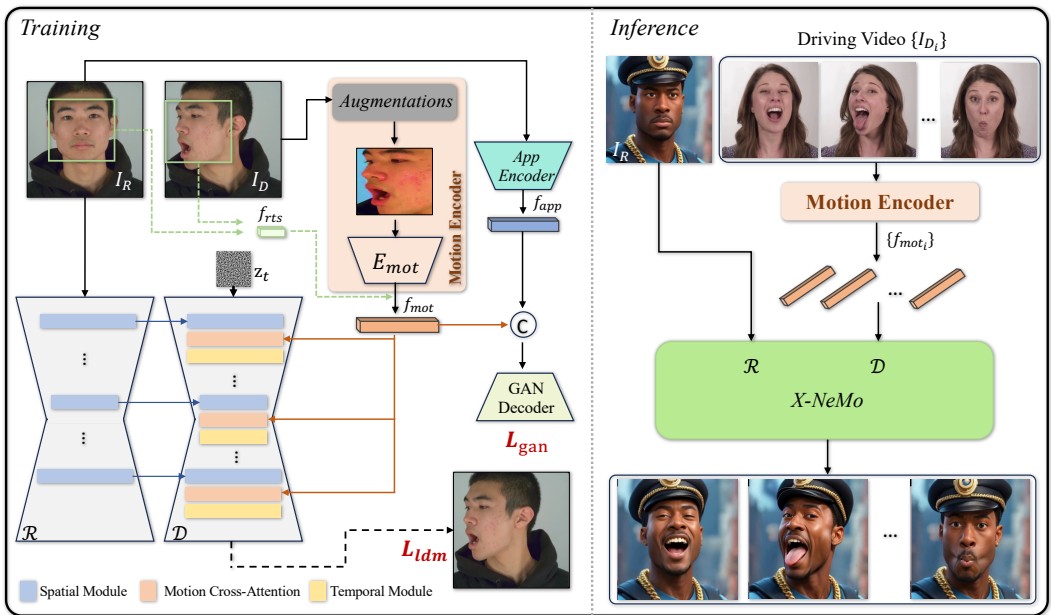

Figure 2: Overview of X-NeMo. We leverage a pretrained diffusion model $\mathcal{D}$ as the rendering backbone and incorporate a reference network module $\mathcal{R}$ for appearance conditioning, along with temporal modules for cross-frame consistency. For motion control, we train a latent motion embedding $f_{mot}$ encoded from the driving image $I_D$ after applying spatial and color augmentations. Alongside the relative translation and scaling $f_{rts}$ of the face bounding box from reference $I_R$ and driving image $I_D$, we integrate the latent motion conditions into the diffusion backbone using newly inserted cross-attention layers. Besides the original diffusion loss $L_{ldm}$, we supervise the learning of our latent motion embedding with a jointly trained GAN decoder head (coupled with an appearance latent embedding $f_{app}$) using image-level losses $L_{gan}$. During inference, we derive the latent motion codes directly from each driving frame, allowing us to synthesize expressive and precise animations while strictly maintain identity resemblance to the reference image.

## 3 METHOD

Given a single portrait as the reference image $I_R$, our objective is to generate a head animation sequence $\{I_{R->D_i}\}$ of length $l$, conditioned on a driving video $I_{D_i}$, where $i = 1, \ldots, l$ denotes the frame index. The generated frames $\{I_{R->D_i}\}$ aim to preserve the identity features and background content depicted in $I_R$ while accurately replicating the head pose and facial expressions featured in each corresponding driving frame $I_{D_i}$. While portrait animation algorithms are generally trained as a frame reconstruction task over video datasets, the $I_R$ and $I_D$ may feature distinct identities during inference, enabling cross-identity motion transfer.

For our task, we harness the generative capabilities of pre-trained Latent Diffusion Models AI (2022) for image generation. Although our method shares some network modules with prior diffusion-based approaches (Section 3.1), it innovates on motion control by addressing the root causes behind the loss of expressiveness and identity resemblance. We introduce our learning framework that achieves fine-grained, identity-agnostic motion control through cross-attention to a co-learned implicit motion descriptor (Section 3.2). To assist the self-supervised learning of motion and identity disentanglement, we present a set of carefully designed training strategies (Section 3.3). Figure 2 provides an overview of our training and inference pipeline.

### 3.1 PRELIMINARIES

**Latent Diffusion Model.** Facilitated by a pretrained auto-encoder, latent diffusion models Rombach et al. (2022) are a class of diffusion models Ho et al. (2020); Song et al. (2020a;b) that synthesize desired samples in the image latent space, starting from Gaussian noise $z_T \sim N(0, 1)$ and refining through $T$ denoising steps. During training, latent representations of images are progressively

corrupted by Gaussian noise $\epsilon$, following the Denoising Diffusion Probabilistic Model (DDPM) framework Ho et al. (2020). A UNet-based denoising backbone network $\mathcal{D}$ containing intervened layers of convolutions and self-/cross-attentions, is trained to learn the reverse denoising process.

**Portrait Animation.** Recently a line of work Tian et al. (2024); Xie et al. (2024); Wei et al. (2024); Xu et al. (2024b); Yang et al. (2024a); Wang et al. (2024); Ma et al. (2024); Chen et al. (2024b) have explored leveraging the generative power of pretrained LDM, such as Stable Diffusion AI (2022), for portrait animation. While exhibiting slight algorithmic variations, these methods generally employ similar components to transfer driving motions onto the reference image. Specifically, a reference network $\mathcal{R}$ Cao et al. (2023), sharing the same architecture with the UNet $\mathcal{D}$, extracts reference features of identity appearance and background which are then cross-queried by the UNet self-attention blocks. Motion control is achieved through an additional module, often formatted in ControlNet Zhang et al. (2023b) or lighter-weight PoseGuider Hu et al. (2023), translating driving conditions into 2D spatially-aligned offsets additive to the UNet features. To maintain consistency across animated frames, temporal modules Guo et al. (2023), which incorporates cross-frame attentions, are intervened with the spatial transformer blocks.

While effective to some extent, prior methods often fall short in expressiveness and suffer from identity leakage in the generated animations. First, expressiveness is limited by the coarse granularity of the driving motion conditions, such as facial landmarks or synthetic training images Xie et al. (2024), which fail to capture complex and subtle expressions like frowning or puckering. Second, while prior approaches mostly address appearance leakage, they overlook the leakage of 2D facial structure and spatial layout embedded in the driving conditions, whether through landmarks or synthesized images. ControlNet-like mechanisms transform these motion conditions into spatially-aligned offsets within the UNet's intermediate features. This reliance on spatial alignment causes the UNet to bypass the need to interpret semantic correspondences between the reference and driving faces, resulting in undesirable identity drift during cross-identity animation at inference.

## 3.2 Pipeline with Identity-Disentangled Implicit Motion Control

As shown in Figure 2, we follow the existing UNet-based latent diffusion framework AI (2022), integrating both the reference network and temporal modules. However, our key innovation lies in a novel motion control module, designed to tackle challenges in motion expressiveness and identity consistency, particularly during cross-identity reenactments. A core design principle of our approach is to distill motion directly from the original driving images, while ensuring the image generation backbone operates independently of any appearance or structural clues from the motion control path.

**Latent Motion Descriptor.** For motion extraction, we employ an image encoder $E_{mot}$, to learn an implicit latent representation, $f_{mot}$, that captures facial motions across varying levels of granularity. Similar to the approaches in Wang et al. (2022; 2023), we formulate the motion latent representation $f_{mot}$ as a *low-dimensional 1-D global* descriptor. The motion encoder $E_{mot}$ consists of intervened layers of convolution-based feature extraction and self-attention, followed by MLP layers, which encode the motion into a 1D latent vector, thereby eliminating spatial positional information (i.e., image structure) along the encoding process. Following the information bottleneck principle Tishby et al. (2000), we employ a larger network capacity (i.e., the reference net $\mathcal{R}$) and higher feature dimensions (i.e., multi-scale feature maps) for appearance modeling, while using a smaller network capacity ($E_{mot}$) and lower feature dimension ($f_{mot}$) for motion encoding. This design, functioning as a low-pass bottleneck filter, encourages the emergence of disentangled representations that effectively capture key semantics of facial motion without entangling with appearance information. Furthermore, unlike previous methods that rely on pretrained motion extractor as the driving conditions (e.g., facial landmarks), our latent motion representation is optimized jointly with the diffusion generator during training. As a result, this allows our model to progressively learn and refine the motion distribution as the diffusion model is trained on more diverse and expressive video data like NerSemble Kirschstein et al. (2023). With that, our approach enhances the expressiveness of the generated animations, as the model adapts to more complex and nuanced facial motions.

**Cross-Attention Control.** To exert motion control on UNet using our latent motion descriptor, one possible approach would be to use a ControlNet-like module to guide the denoising process after transforming the 1D latent code $f_{mot}$ into a 2D spatially-aligned control map via a StyleGAN-like decoder. However, this would contradict our design goal for identity disentanglement.(Figure 3(a)) Since $f_{mot}$ is intentionally free of 2D structural information, transforming it into a spatial control

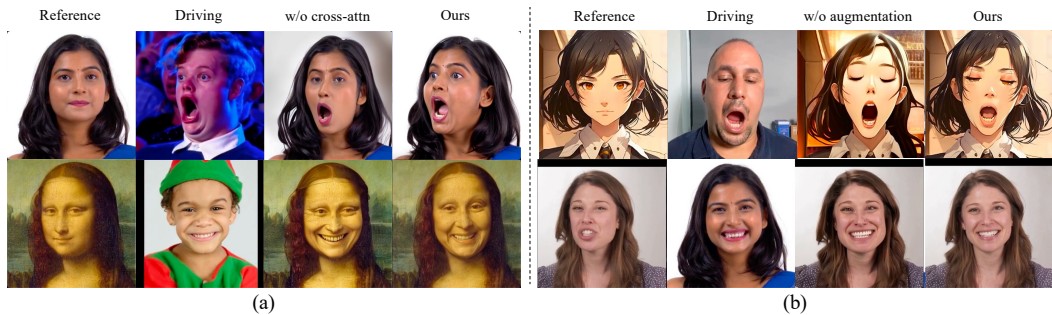

Figure 3: Qualitative ablation study on factors affecting identity consistency. (a) Replacing our motion cross-attentions with a control module using spatially additive guidance leads to severe leakage of the driving identity's facial structure. (b) Training without our color and spatial augmentations results in noticeable appearance leakage and identity drift.

map demands additional input regarding the reference identity's structure, thereby violating our principle that the motion control path should remain agnostic to identity-specific features. Instead the UNet should resort to the reference net for relevant identity-related information.

Instead, we adopt a cross-attention conditioning mechanism, which has proven effective across various control modalities Rombach et al. (2022); Tian et al. (2024); Ruiz et al. (2023). This allows direct injection of the latent motion embedding into the UNet without adding spatial bias. Specifically, we insert motion-attention layers performing cross attention with the latent motion code $f_{mot}$ after each spatial transformer blocks in the backbone. This cross-attention scheme integrates the 1D motion embedding globally into the generation process, encouraging the UNet to interpret the motion condition and establish semantic correspondences between the reference and driving identity.

### 3.3 TRAINING STRATEGIES

For training, we randomly sample two distinct frames from a video as the reference $I_R$ and driving $I_D$ image, respectively. The model is then trained to denoise the latent map of the target image $I_D$ at timestep $t$, with the diffusion loss defined as follows,

$$L_{ldm} = \mathbb{E}_{z_t, \epsilon \sim \mathcal{N}(0,1), t} \left[ \| \epsilon - \epsilon_\theta(z_t, c_{ref}, f_{mot}) \|_2^2 \right], \tag{1}$$

where $\epsilon_\theta$ denotes the trainable parameters in the backbone $\mathcal{D}$, and $f_{mot}$, $c_{ref}$ represent the driving motion and reference features, respectively, extracted by $E_{mot}$ and $\mathcal{R}$. The training process is structured into three stages. The first stage is the image pretraining stage, where the backbone UNet and reference net are taken into training. The second stage additionally incorporates the motion encoder $E_{mot}$ and the newly integrated motion-attention layers into the optimization, forming an end-to-end encoder-generator structure. Lastly, we train the temporal modules to ensure cross-frame coherence. After the three-stage training process (intended as warm-up), we finetune the entire pipeline end-to-end, with all trainable modules optimized simultaneously.

However, straightforward self-supervised training of the entire framework does not inherently disentangle identity from facial motion. The UNet may inadvertently reconstruct the target image by borrowing appearance features from the driving image or encoding identity information into the latent motion descriptor $f_{mot}$. Additionally, when $I_R$ and $I_D$ share similar expressions, the model may distill motion signals from the reference image , hindering independent control over facial motion and identity, particularly in cross-identity reenactments. To address these issues, we propose several training strategies to fully leverage the potential of our network design in Section 3.2.

**Color/Spatial Augmentation.** To suppress identity information leakage from the motion control branch, we reduce the appearance and structural consistency between the driving and target images using both color and spatial augmentations. Specifically, we apply color jittering, random scaling within 30%, and piecewise affine transformations, to the driving image $I_D$, altering the facial appearance and shape while preserving motion semantics. We also perform face-centered cropping to enhance spatial disparity between the driving and target, promoting the motion encoder $E_{mot}$ to focus on the facial movements and capture nuanced expressions. As shown in Figure 3(b), these augmentations effectively guide the motion encoder $E_{mot}$ towards learning identity-agnostic mo-

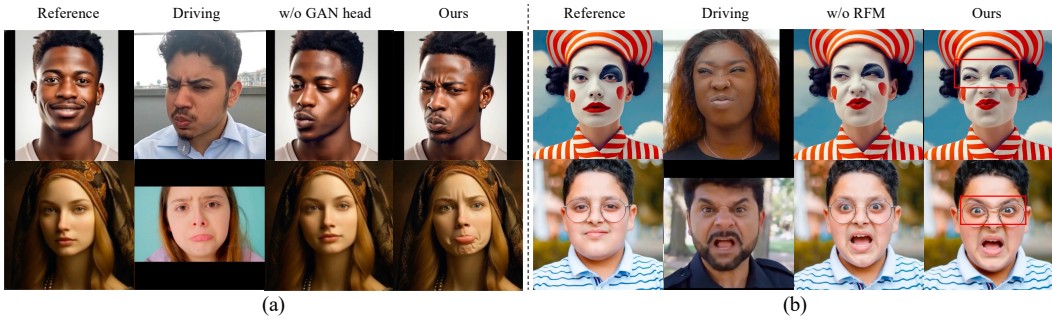

Figure 4: Qualitative ablation study on factors influencing motion expressiveness.(a) Without the dual GAN head, training solely with the diffusion loss hinders the motion encoder's ability to learn detailed and local motion patterns. (b) Our reference feature masking (RFM) strategy facilitates the transfer of fine-level facial expressions, such as the wrinkles at the nasal region.

tion representation. To account for the disrupted head translation due to face-centered cropping, we construct a triplet $f_{rts} = (\Delta x/s_r, \Delta y/s_r, s_d/s_r)$, where $(\Delta x, \Delta y)$ denotes the 2D relative distance between the face centers in the $I_R$ and $I_D$, and $s_d/s_r$ reflects the change in bounding box scale. This triplet is processed through fully connected layers and fused with latent motion embedding $f_{mot}$.

Owing to our design of motion control via latent motion embedding and cross-attentions, we substantially improves facial motion disentanglement from identity structure through spatial augmentations. In contrast, applying these augmentations with a ControlNet-like mechanism, which relies heavily on aligned spatial control signals, would degrade both robustness and accuracy.

**Dual-Head Latent Supervision.** In our early experiments, we observe that our motion control training, while effective in capturing coarse facial motions, converges slowly and struggles to depict subtle and fine-grained motions like frowning and puckering (see Figure 4(a)). The latent motion embedding, acting as a low-pass filter, tends to model low-frequency movements first. Additionally, the diffusion loss used during training assigns equal weight to every "pixel" in the latent noise, leading the model to prioritize a smooth motion space over capturing local, detailed expressions. To address this, we introduce a dual GAN-based head to guide the learning of the latent motion embedding, enhancing the model's attention to fine-grained facial expressions.

Following Burkov et al. (2020); Wang et al. (2022; 2023), we employ a convolutional feature extractor network to encode the reference image into an appearance latent embedding $f_{app}$. Together with our motion latent code $f_{mot}$, these embeddings modulate a StyleGAN generator Karras et al. (2020) (i.e., the GAN-head decoder) to generate an RGB image, co-trained with our diffusion-based motion control. Its training losses, collectively denoted as $L_{gan}$, are formulated in image space, including a weighted $L_1$ reconstruction loss, adversarial loss, feature matching loss Burkov et al. (2020), and VGG perceptual losses Simonyan & Zisserman (2014); Cao et al. (2018). Focused on structural variations more than pixel-wise differences, these image-level losses guide the latent space learning with detailed and local motion modes. Furthermore, since the GAN head contains much fewer trainable parameters than the diffusion backbone, it converges faster, boostrapping the motion encoder and aiding the learning of motion attention layers under a well-distributed motion latent embedding.

**Reference Feature Masking.** In line with our strategies for identity disentanglement in the motion control branch, we also aim to mitigate motion leakage from the appearance reference network. When $I_R$ and $I_D$ exhibit similar expressions, even just partially, the backbone network is likely to utilize the high-dimensional multi-scale appearance features as a "shortcut" for motion reference, bypassing the intended reliance on our compact 1D latent motion descriptor. While such motion leakage does not impede training data fitting during self-driven training, it hampers the learning of effective and expressive motion control (Figure 4(b)).

Inspired by Masked Image Modeling He et al. (2022), we introduce reference feature masking to mitigate motion leakage in appearance features. Specifically, we apply 30% uniform random masking to the appearance feature maps from the reference net $\mathcal{R}$. The masked feature maps are flattened and used as reference keys and values for the self-attention layers within the UNet backbone. This balances the strength between appearance and motion signals, ensuring that subtle driving expressions are effectively transferred without being overshadowed by the reference expressions.

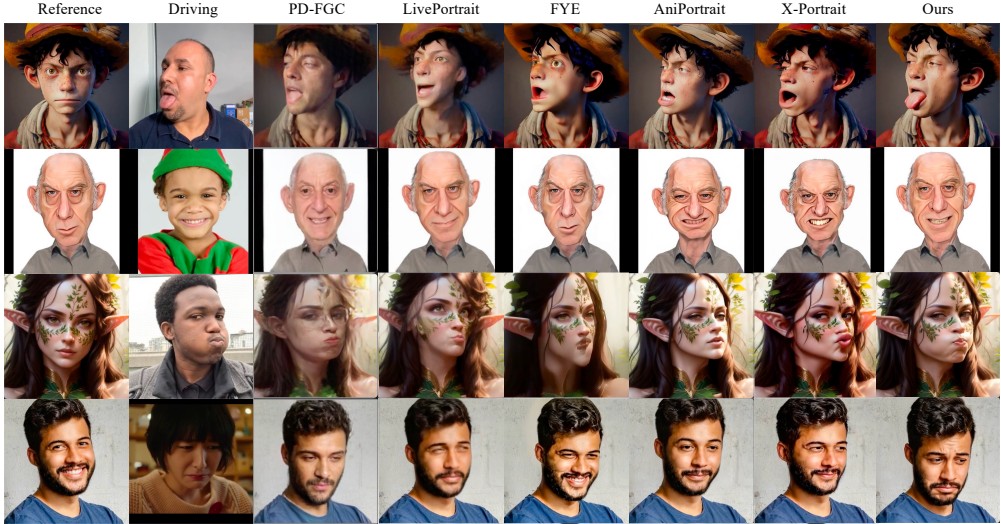

Figure 5: Qualitative comparisons. Among all the methods, X-NeMo achieves the most accurate transfer of intricate expressions and emotional subtleties while demonstrating the highest identity resemblance, regardless of the characteristic differences between the reference and driving identities.

## 4 EXPERIMENT

### 4.1 IMPLEMENTATION DETAILS

We train our model on a combination of talking head datasets (HDTF Zhang et al. (2021), VFHQ Xie et al. (2022)) and facial expression dataset (NerSemble Kirschstein et al. (2023)), uniformly processed at 25 fps and cropped to a $512 \times 512$ resolution. The training is conducted on 8 Nvidia A100 GPUs using the AdamW optimizer Yao et al. (2021) with a learning rate of $1e - 5$. We use a batch size of 64 for appearance and motion control training, and a batch size of 16 for the temporal module using 24-frame video sequences. During inference, we implement the prompt traveling technique Tseng et al. (2022) to enhance temporal smoothness in long video generation.

For evaluation, we compile a benchmark of 100 in-the-wild reference portraits DeviantArt (2024); Midjourney (2024); Pexels (2024), representing a broad spectrum of facial structures, appearances and styles. Additionally we collect 100 test videos from DFEW Jiang et al. (2020) featuring emotionally expressive clips, alongside 200 licensed videos showcasing a diverse range of emotions, head poses, and facial expressions. Please also refer to our supplementary video for more results.

### 4.2 EVALUATIONS AND COMPARISONS

In our evaluation, we compare our method against state-of-the-art video-driven portrait animation baselines, including X-Portrait Xie et al. (2024), AniPortrait Wei et al. (2024), Follow-your-Emoji (FYE) Ma et al. (2024), and Echomimic Chen et al. (2024b). We also assess recent non-diffusion-based methods, including PD-FGC Wang et al. (2023) that employs latent motion representation, and LivePortrait Guo et al. (2024a) which uses implicit neural landmarks. EmoPortraits Drobyshev et al. (2024), a GAN-based expressive portrait animation method, is excluded from our comparisons due to the lack of inference code. For fair comparisons, we finetune AniPortrait, X-Portrait, and PD-FGC using our dataset, while utilizing the released pretrained models for the remaining methods, given the unavailability of their training code. We assess performance in both self and cross reenactments, with all metrics computed at a resolution of $256 \times 256$ (the resolution at which PD-FGC was trained).

**Self Reenactment.** For each test video, we utilize the first frame as the reference image, generating the entire sequence with subsequent frames acting as both the driving image and the ground truth target. We evaluate the performance by computing the L1, structural (SSIM), and perceptual (LPIPS) image losses to assess both image quality and motion accuracy. Our method, X-NeMo, consistently outperforms all baseline methods, as shown in our numerical comparisons (Table 1).

Table 1: Quantitative comparison. Our method achieves superior numerical results than all the baselines in both self-driven and cross-driven reenactments.

| Method | Self-Reenactment | | | Cross-Reenactment | | | |
|---|---|---|---|---|---|---|---|
| | L1↓ | SSIM↑ | LPIPS↓ | ID-SIM↑ | AED/APD↓ | EMO-SIM↑ | FVD↓ |
| PD-FGC | 0.085 | 0.728 | 0.291 | 0.604 | 0.045/3.95 | 0.49 | 441.8 |
| LivePortrait | 0.074 | 0.770 | 0.236 | 0.702 | 0.055/6.61 | 0.48 | 279.2 |
| X-Portrait | 0.063 | 0.793 | 0.209 | 0.695 | 0.041/4.07 | 0.52 | 237.3 |
| FYE | 0.075 | 0.741 | 0.249 | 0.725 | 0.062/4.49 | 0.41 | 360.9 |
| AniPortrait | 0.057 | 0.812 | 0.198 | 0.713 | 0.043/4.14 | 0.46 | 290.5 |
| Ours | **0.055** | **0.826** | **0.168** | **0.787** | **0.039/3.42** | **0.65** | **152.8** |

**Cross Reenactment.** Our method empowers the creation of captivating and expressive animations across diverse portraits even driven by in-the-wild videos with distinct identity features (Figure 1, 3, 4, 5). Our qualitative comparisons (Figure 5) demonstrate that X-NeMo surpasses all the baselines by a significant margin in identity similarity, expression accuracy and perceptual quality. GAN-based baselines suffer from blurriness and distortions under large head motions and when applied to out-of-domain portraits. For both exaggerated (e.g., cheek puffing, sticking out the tongue) and subtle facial expressions (e.g., biting the lip), all other methods struggle to faithfully capture and transfer these facial motion details. Additionally, our method excels in preserving identity resemblance, regardless of the structural difference between the reference and driving faces, while severe identity drift occurs in other diffusion-based baselines relying on spatially aligned control signals.

For quantitative assessment, video-level evaluation uses FVD Unterthiner et al. (2018) metrics. To evaluate the image-level quality, we employ three metrics to evaluate identity similarity, expression/head pose accuracy, and emotion consistency respectively. Specifically, we utilize the ArcFace score Deng et al. (2019) to measure the cosine similarity of identity features (ID-SIM). Motion accuracy is calculated as the average $L_1$ difference between extracted facial blendshapes (AED) and head poses (APD) of the driving and generated images using MediaPipe Lugaresi et al. (2019). However, since blendshapes provide only a coarse motion estimation, we further employ a pretrained emotion encoder, EmoNet Toisoul et al. (2021), to assess emotion accuracy. Specifically, we calculate the mean value of concordance correlation coefficients and Pearson correlation coefficients for both valence and arousal to measure the emotion similarity (EMO-SIM). The emotion score reflect model's performance in fine-grained expression control, as emotion recognition is highly sensitive to micro-expressions. Our method numerically surpasses all competitors, demonstrating the superior capabilities afforded by our novel motion control design (Table 1).

**Applications.** Our latent motion descriptor acts as a unified representation for both motion comprehension and generation, supporting tasks beyond portrait animation, including (emotion-conditioned) portrait video outpainting and latent motion interpolation. With our expressive, identity-agnostic motion embedding, we are able to generate long-range expressive videos while consistently preserving the identity across diverse portraits. For more details and visual results, please refer to our supplemental paper (Section. E) and accompanying video.

## 4.3 Ablation Studies

We ablate individual design choices by removing them from our full training pipeline. We validate the function of dual-head supervision in motion expressiveness by removing the GAN decoder from co-training ("w/o GAN head"). Even with extended training, the motion

Table 2: Quantitative ablation.

| Method | ID-SIM↑ | AED/APD↓ | EMO-SIM↑ |
|---|---|---|---|
| w/o GAN head | 0.789 | 0.045/4.64 | 0.43 |
| w/o joint training | 0.782 | 0.040/3.49 | 0.52 |
| w/o RFM | **0.791** | **0.039/3.41** | 0.62 |
| w/o augmentation | 0.724 | 0.042/3.63 | 0.50 |
| w/o cross-attn | 0.697 | 0.040/3.55 | 0.48 |
| Ours | 0.787 | **0.039**/3.42 | **0.65** |

encoder $E_{mot}$ struggles with detailed motions in the absence of image-level loss guidance ( Figure 4(a)), as reflected by a substantial reduction in both expression and emotion metrics (Table 2). We further validate the importance of joint training by pretraining $E_{mot}$ with GAN losses and then freezing it while training the rest of the model sorely with the diffusion loss ("w/o joint training"). Both Table 2 and the supplemental video demonstrate that joint training elicits stronger motion

representation from the encoder, leveraging the diffusion model's superior generative capacity over the standalone GAN decoder. Additionally, we assess the role of reference feature masking ("w/o RFM") in enhancing motion accuracy. Without it, the network shows a stronger bias to the reference expressions at certain local regions(Figure 4(b)), yielding a lower emotion score (Table 2).

We assess the efficacy of augmentation and cross-attention control in motion-identity disentanglement. When training without augmentations ("w/o augmentation"), the generations often exhibit noticeable identity leakage from the driving subject in both appearance and face structure (Figure 3(b)), as confirmed by the drop in identity similarity score (Table 2). We also compare our method to a baseline where the motion latent is transformed into a 2D control map via an upsampling decoder and applied to the UNet using a ControlNet ("w/o cross-attn"). While effective for coarse motion control, its reliance on spatially-aligned additive controls lead to reduced identity resemblance (Figure 4(a)), underscoring the necessity of our structure-agnostic motion control design.

We leverage classifier-free guidance (CFG) Ho & Salimans (2022) to steer the inference towards more expressive motion transfer. While straightforward for the conditional generation, we find the optimal practice by using fully masked appearance features and the motion latent $f_{ref\_mot}$ extracted from reference image $I_R$ as the negative prompts. As illustrated in Figure 6, this CFG configuration enables the network to better distinguish between conditional appearance and

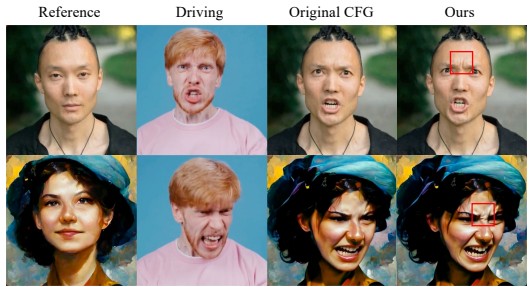

Figure 6: Ablation on different CFG configurations.

motion features, facilitating more accurate and semantic motion transfer. Our CFG is formulated as

$$\tilde{\epsilon}_\theta(z_t, c_{ref}, f_{mot}) = (1 + w)\epsilon_\theta(z_t, c_{ref}, f_{mot}) - w\epsilon_\theta(z_t, \emptyset, f_{ref\_mot}), \tag{2}$$

where $w = 3.5$ is the CFG scale and $\tilde{\epsilon}_\theta$ is the final composed noise estimate.

## 5 DISCUSSION AND CONCLUSION

We present X-NeMo, a novel diffusion-based portrait animation framework that effectively disentangles motion and identity, achieving substantial improvements in generating expressive, identity-preserved animations from diverse portraits. At its core, we introduce self-supervised learning framework that integrates latent motion representations with structure-agnostic motion control through cross-attentions, enhanced by carefully-designed training and inference strategies. We demonstrate high-quality animation results on a wide range of portraits and expressive driving videos, validating the efficacy of our approach. We believe our method offers valuable insights into the field and opens avenues for numerous downstream tasks.

**Limitations and Future Work.** Our method is trained solely on real human talking and expression videos. Consequently, out-of-domain portraits with non-human appearances, such as 3D cartoon characters, may exhibit artifacts like blurred eyes. Additionally, it might strug-

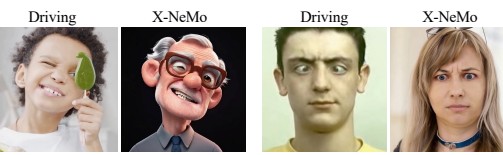

Figure 7: Failure cases.

gle with exaggerated expressions absent from the training data (see Figure 7). However, our scalable framework, free from reliance on pre-trained motion detectors, enables generalization to various styles and motions with more training data. Our motion control represents a general scheme, with which we aim to integrate into video diffusion backbones Yang et al. (2024b); Zheng et al. (2024) in the future, for smoother and more dynamic results.

**Ethics Statement.** Our work aims to improve portrait animation from a technical perspective and is not intended for malicious use. However, we recognize the potential for misuse like generating fake videos. Therefore, synthesized images and videos should clearly indicate their artificial nature.

**ACKNOWLEDGMENTS**  This paper is supported by the NSFC project No.62125107.

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

## A  TRAINING AND INFERENCE DETAILS

**GAN Head Training Losses.**  Following Burkov et al. (2020), we train our dual GAN decoder in a self-supervised manner to reconstruct $I_D$ using a combination of losses. Specifically, a $L_1$ reconstruction loss is employed to minimize pixel-wise $L1$ distance:

$$\mathcal{L}_{recon} = \|I_D - I_{R \to D}\|_1 \tag{3}$$

Additionally, two perceptual losses, $L_{VGG}$ and $L_{VGGFace}$, are applied based on $L_1$ matching of ConvNet activations from a VGG-19 model Simonyan & Zisserman (2014) pretrained for ImageNet classification and a VGGFace model Cao et al. (2018) trained for face recognition:

$$\mathcal{L}_{vgg} = \sum_{i=1}^{N} \|\text{VGG}^i(I_D) - \text{VGG}^i(I_{R \to D})\|_1, \tag{4}$$

$$\mathcal{L}_{vggf} = \sum_{I=1}^{N} \|\text{VGG}_{\text{face}}{}^i(I_D) - \text{VGG}_{\text{face}}{}^i(I_{R \to D})\|_1, \tag{5}$$

where $N$ denotes the number of feature layers in each respective pre-trained VGG model. An adversarial generative loss $L_{adv}$ is applied using a co-trained discriminator D, while a feature matching loss $L_{fm}$ is calculated as the $L_1$ distance between discriminator feature maps at different layers:

$$\mathcal{L}_{fm} = \sum_{i=1}^{N} \|\text{D}^i(I_D) - \text{D}^i(I_{R \to D})\|_1 \tag{6}$$

The overall learning objective for the GAN head is then formulated as:

$$\mathcal{L}_{gan} = \mathcal{L}_{\text{adv}} + \lambda_r \mathcal{L}_{recon} + \lambda_{vgg} \mathcal{L}_{\text{vgg}} + \lambda_{vggf} \mathcal{L}_{\text{vggf}} + \lambda_{fm} \mathcal{L}_{fm} \tag{7}$$

where $\lambda_r$=1.0, $\lambda_{vgg}$=3e-2, $\lambda_{vggf}$=6e-3 and $\lambda_{fm}$=10.0.

**Inference Performance.**  During inference, we use 25 DDIM steps Song et al. (2020a) with a classifier-free guidance (CFG) scale of 3.5. For generating a 1-second video at 25 frames per second, the process takes approximately 20 seconds and requires 24 GB of memory.

**Encoder Architecture.**  $E_{mot}$ takes the classical feature alignment network Bulat & Tzimiropoulos (2017) as the backbone, with an additional attention layer added at both its input and output to enhance feature extraction capabilities. Finally, it outputs a 1D vector through two MLP layers. The appearance encoder for extracting $f_{app}$ is implemented as a ResNet50.

## B  VIDEO-LEVEL PERFORMANCE ANALYSIS

As demonstrated in our comprehensive quantitative (Table 1) and qualitative evaluations(Figure 5 and video demos), X-NeMo consistently outperforms all baseline methods in motion expressiveness and temporal coherence. This strong video-level performance stems primarily from our robust single-frame reenactment capability. Specifically, our motion encoder captures fine-scale facial motion within individual frames, while the temporal motion module focuses on cross-frame consistency, supported by the SVD-VAE decoder Blattmann et al. (2023) and prompt traveling Cao et al. (2023) during inference.

Unlike prior methods, which rely on pre-trained motion control signals such as explicit landmarks (AniPortrait Wei et al. (2024)) or implicit landmarks (X-Portrait Xie et al. (2024)), our approach utilizes a learnable motion representation that is optimized jointly with the diffusion generator. Pre-trained motion signals are often unstable and prone to jitter across frames, and they fail to fully capture the complex facial expressions present in diverse datasets. By jointly optimizing motion representation with the generator, X-NeMo inherently achieves better expressiveness and more effective disentanglement of motion and identity.

However, straightforward self-supervised training does not naturally disentangle identity from facial motion. As shown in Table 2, critical components like Dual Head Supervision, Cross-Attention, and Augmentations are essential for achieving effective disentanglement. Furthermore, Joint Training and RFM enhance motion extraction, capturing subtle micro-expressions (not captured by AKD but reflected in the EMO-SIM metric) and enabling vivid and expressive reeactment results.

## C  MORE ABLATIONS

We provide additional visual ablations on some network and training hyperparameters.

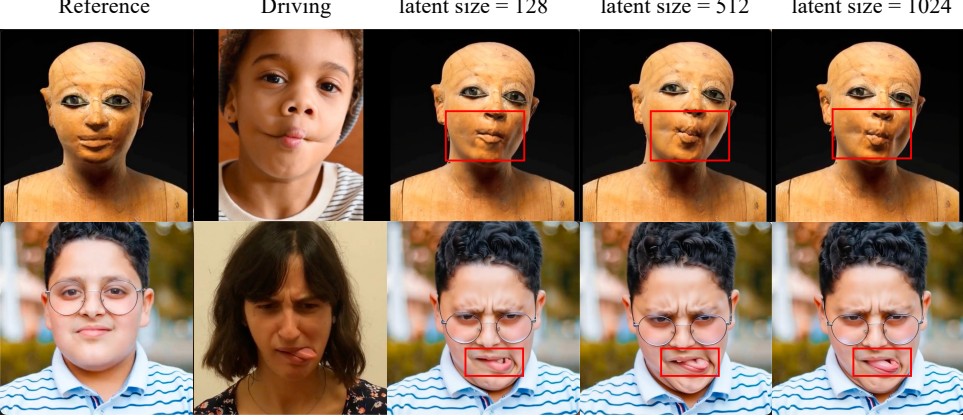

Figure 8: We ablate the effect of different sizes of latent motion embedding in capturing fine-grained intricate expressions.

**Motion Latent Embedding Size.**    We evaluate the dimensionality of our motion latent embedding by comparing 128, 512, and 1024-dimensional latent codes for $f_{mot}$. In typical driving scenarios, all three configurations perform similarly in replicating facial expressions with minimal differences. However, as shown in Figure 8, reducing the embedding size to 128 diminishes the ability to capture subtle, intricate expressions, while increasing it to 1024 provides negligible improvements. Thus, we select the 512-dimensional embedding as it balances compactness with motion expressiveness.

**Reference Feature Masking Ratio.**    We assess the effectiveness of our reference feature masking strategy across different masking ratios, ranging from 0%, 30%, 75%, to 95%. As shown in Figure 9, this strategy enhances the transfer of detailed facial expressions; however, excessively high masking ratios impede the model's ability to capture fine motion details and maintain identity consistency.

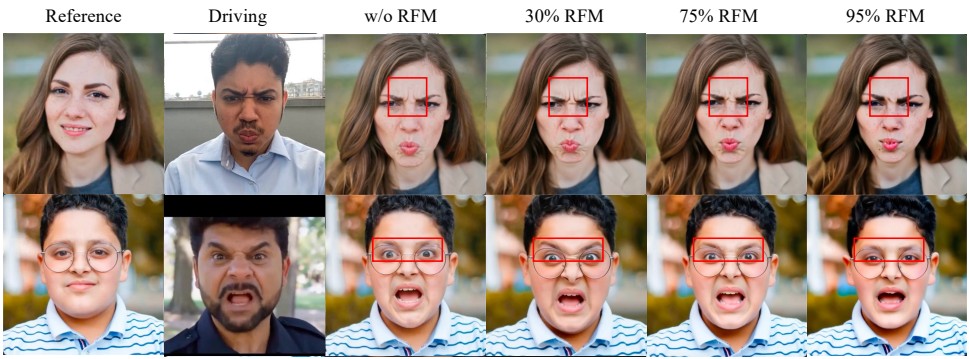

Figure 9: Qualitative comparison of different ratios of reference feature masking indicates that 30% achieves the most accurate capture of driving facial motions.

Table 3: More quantitative ablations.

| Method | ID-SIM↑ | AED/APD↓ | EMO-SIM↑ | FVD↓ |
|---|---|---|---|---|
| Ours | **0.787** | **0.039/3.42** | **0.6531** | 152.8 |
| Temporal motion encoder | 0.786 | 0.039/3.42 | 0.6526 | **151.5** |
| Simultaneous training | 0.774 | 0.039/3.45 | 0.6398 | 155.2 |

This is likely because when the reference image's appearance is too heavily obscured during training, the motion encoder compensates by encoding appearance information, reducing the capacity of the motion embedding for expressing dynamic movements. In practice, we found that masking ratios between 20% and 50% achieve optimal results, with 30% used in our implementation.

**Single-Frame Motion Extraction**  We conducted an experiment where latent motion sequences were extracted using multiple driving frames instead of a single image. Specifically, we implemented a pipeline with a temporal motion encoder trained end-to-end using 16-frame video clips. As shown in Tab. 3, the differences across all metrics are minimal. However, training with video clips significantly reduces training efficiency, as the batch size for multi-frame training is reduced by a factor of 16 compared to single-frame training. Consequently, we opted for the more efficient single-frame pipeline.

Regarding why temporal encoding yields negligible gains for portrait animation, we believe this is due to the structured nature of facial video data and the capability of our motion encoder, which learns directly and implicitly from smooth RGB image signals. Facial videos exhibit strong structural consistency and clear motion patterns, enabling fine-grained motion extraction on a frame-by-frame basis. This ensures temporal continuity without requiring multi-frame inputs. Additionally, in our self-driven training setup, the diffusion generation network accurately reconstructs the target image using a single driving frame, often disregarding motion cues from neighboring frames.

**Simultaneous Training**  As mentioned in Sec. 3.3, before the end-to-end fine-tuning, our method conducts a three-stage training process. To ablate this, we implemented a baseline that directly trains the whole network simultaneously. As shown in Tab. 3, the differences across all metrics are not remarkable. Nonetheless, due to the high GPU cost of simultaneous training and small batch size, it's much slower for this baseline to converge. Hence in practice, we adopt the stepwise training strategy for training efficiency.

# D    MORE RESULTS

Please refer to our supplemental video for more expressive demo cases.

# E  APPLICATIONS

Our latent motion embedding, trained jointly with the diffusion backbone, offers a compact, identity-agnostic, yet expressive representation for a diverse range of facial motions. Beyond the primary portrait animation task, we showcase its broader applications as a unified motion representation, enabling seamless motion interpolation, video outpainting and conditioned generation.

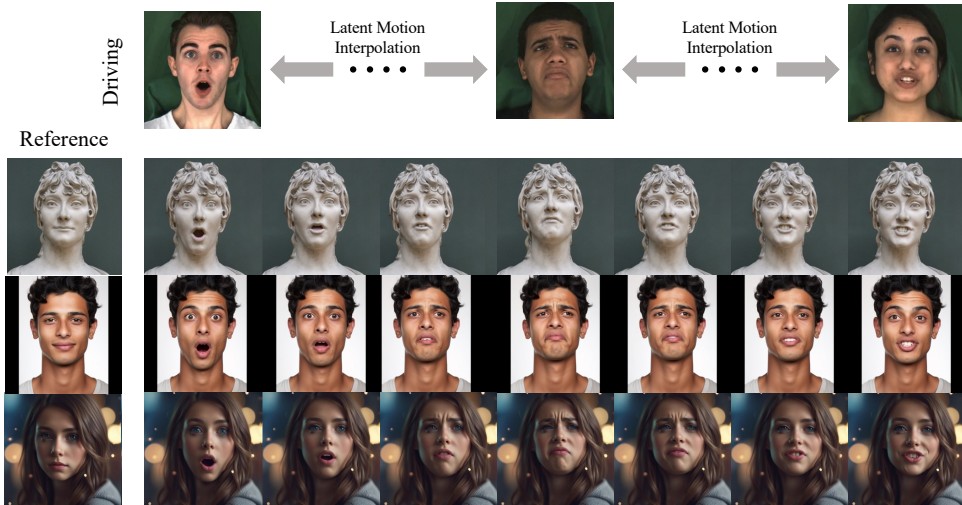

Figure 10: Latent motion interpolation. We derive latent motion codes from a few driving keyframes (top) and apply onto diverse portraits with linearly interpolated motion embeddings (bottom).

**Latent Motion Interpolation.**   Owing to our smooth and identity-agnostic latent motion space, we are able to extract keyframe expressions from different videos, linearly interpolate the latent embeddings and apply them across diverse portraits, as showcased in Figure. 10. This interpolation yields smooth and natural expression transitions, maintaining motion coherence across different portraits and appearance consistency with the reference images. These results underscore the robustness and identity disentanglement of our motion latent embedding.

**Portrait Video Outpainting and Generation.**   By leveraging our motion latent embedding as a unified representation for motion comprehension and generation, we showcase its application in video outpainting. Specifically, we adopt the approach from T2M-GPT Zhang et al. (2023a) to tokenize temporal latent motions by training a Vector-Quantized VAE model Esser et al. (2021) with a learnable codebook (4096 entries of 8-dimension code) that downsamples the temporal dimension by a factor of 4. This allows us to represent $T$ frames of motion with $T/4$ discrete motion tokens, where $T$ is the training sequence length (we use $T = 128$), facilitating the use of GPT-like frameworks for long-sequence motion generation. In Figure. 12, we train a GPT2-small network that extends preceding motions derived from a driving video with extrapolated motions. The results show natural and expressive generated sequences, thanks to the strong representation power of our latent motion embedding. Moreover, as more facial video datasets containing multimodal annotations (e.g., text and audio) become available, our method can seamlessly extend to multimodal facial video generation within a unified framework. As an example, we illustrate emotion-conditioned portrait video generation in Figure 11, trained with MEAD dataset Wang et al. (2020).

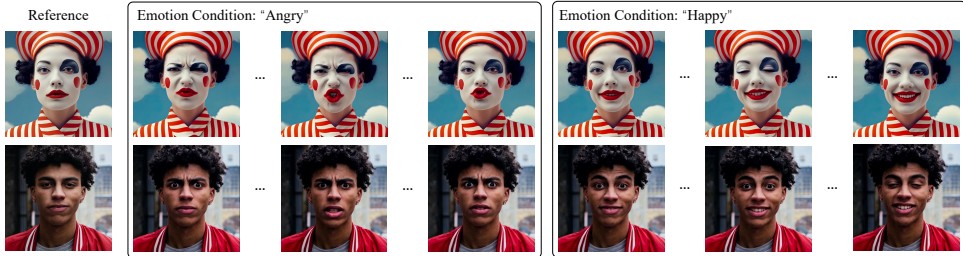

Figure 11: Emotion-conditioned portrait video generation.

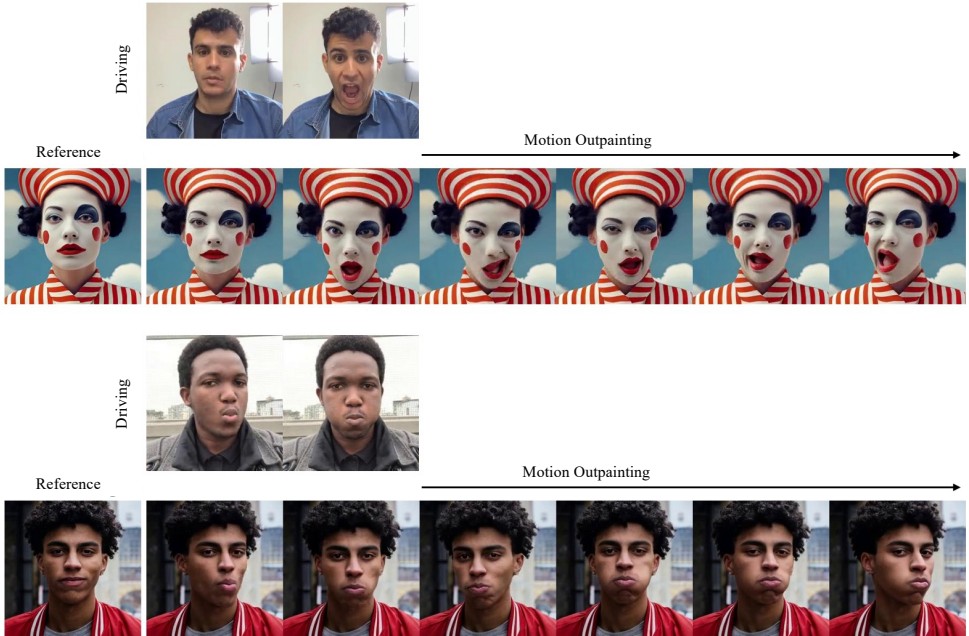

Figure 12: Portrait video outpainting. Starting from a sequence of driving motion, our model is capable of extrapolating into a long video sequence with consistent identity attributes.

