# OpenReview forum: "X-NeMo: Expressive Neural Motion Reenactment via Disentangled Latent Attention"
_ICLR.cc/2025/Conference — ICLR 2025 Poster_

### Official Review · Reviewer_tmTm · 2024-10-28

**Soundness:** 2
**Presentation:** 3
**Contribution:** 3
**Rating:** 6
**Confidence:** 4

**Summary:**

The paper introduces X-NeMo, a novel zero-shot, diffusion-based portrait animation framework that animates static portraits using head movements and facial expressions from a driving video of another subject. The authors identify key challenges in existing approaches, such as identity leakage and difficulty in capturing subtle and extreme facial expressions. To overcome these issues, X-NeMo employs a fully end-to-end training process that extracts a 1D identity-agnostic motion descriptor from the driving image, controlling the motion in the generated animation through cross-attention rather than traditional spatial guidance. This technique mitigates the transmission of identity clues from the driving video, reducing identity leakage and improving expressiveness. X-NeMo learns facial motion from diverse video datasets without relying on pre-trained motion detectors, using a dual GAN-based decoder and various augmentations to disentangle motion from identity cues. By embedding motion in a 1D latent vector and leveraging cross-attention, the model avoids transferring spatial structures, thereby preserving identity resemblance in the animated portraits. Experiments demonstrate that X-NeMo surpasses state-of-the-art methods, producing highly expressive animations while maintaining the subject’s identity.

**Strengths:**

1.	The design of the implicit 1D latent motion descriptor and its integration through cross-attention offers a perspective on addressing identity leakage and expressiveness, overcoming the shortcomings of explicit motion descriptors.
2.	X-NeMO better captures subtle and extreme expressions in the process of portrait animation compared to previous models.

**Weaknesses:**

1.	X-NeMo addresses the portrait animation task in the image level, and derive latent motion codes from each driving frame without the perception of the whole driving video. This setting neglect the video continuity, and may not capture the subtle coherence of expressions in a driving video.

**Questions:**

None

---

> ### Author Response · Authors · 2024-11-19
> **Response to Reviewer tmTm**
>
> Thank you for your insightful comments and appreciation of the effectiveness of our method.
>
> **Re:  Temporal v.s. Single-Frame Motion Extraction**
>
> Thank you for your thoughtful question. We conducted similar experiments where latent motion sequences were extracted using multiple driving frames instead of a single image. Specifically, we implemented a pipeline with a temporal motion encoder trained end-to-end using 16-frame video clips. The results of this ablation study are summarized below:
>
> | Method | ID-SIM$\uparrow$ | AED/APD$\downarrow$ | EMO-SIM$\uparrow$ | FVD$\downarrow$ |
> |------------------|--------|---------|---------|-----|
> | Ours | 0.787 | 0.039/3.42 | 0.6531 | 152.8 |
> | Temporal motion encoder | 0.786 | 0.039/3.42 | 0.6526 | 151.5 |
>
> As shown in the table, the differences across all metrics are minimal. However, training with video clips significantly reduces training efficiency, as the batch size for multi-frame training is reduced by a factor of 16 compared to single-frame training. Consequently, we opted for the more efficient single-frame pipeline.
>
> Regarding why temporal encoding yields negligible gains for portrait animation, we believe this is due to the structured nature of facial video data and the capability of our motion encoder, which learns directly and implicitly from smooth RGB image signals. Facial videos exhibit strong structural consistency and clear motion patterns, enabling fine-grained motion extraction on a frame-by-frame basis. This ensures temporal continuity without requiring multi-frame inputs. Additionally, in our self-driven training setup, the diffusion generation network accurately reconstructs the target image using a single driving frame, often disregarding motion cues from neighboring frames.
>
> While temporal motion extraction may not provide significant benefits for our current task, we recognize its potential for future applications. We aim to explore this design further, particularly for generating dynamic temporal effects (e.g., simulating hair motion) or addressing more complex video datasets, such as full-body driving tasks.

---

> ### Author Response · Authors · 2024-11-25
> **Have our responses addressed your concerns？**
>
> Dear Reviewer tmTm,
>
> We thank you for your helpful comments on our paper. We have taken your concerns into consideration and addressed them carefully. We have also conducted additional experiments to support our claims, and we have included our responses in our rebuttal. We hope that our rebuttal has convinced you. Has this response clarified our method and experiments? We kindly ask the reviewer to reassess the paper in light of our response, if it clears things up. If not, we would be happy to provide alternative explanations, or to provide clarifications on any other part of the paper. Thanks again for your time reviewing our paper.

---

### Official Review · Reviewer_qbkt · 2024-11-02

**Soundness:** 2
**Presentation:** 2
**Contribution:** 2
**Rating:** 3
**Confidence:** 2

**Summary:**

The paper proposes X-NeMo, a diffusion-based portrait animation framework. It tries to address identity leakage and capturing diverse expressions. The method involves extracting a 1D identity-agnostic latent motion descriptor from the driving image, using cross-attention for motion control in image generation. It is trained end-to-end with a dual GAN decoder and spatial/color augmentations.

**Strengths:**

+ The paper is well-structured.
+ The problem of portrait animation with high expressiveness and identity preservation is important.
+ The use of a 1D latent motion descriptor and cross-attention for motion control is reasonable.

**Weaknesses:**

- What is the definition of zero-shot here? Firstly the model is trained. Secondly in the inference several reference images are provided. Thirdly the description of zero-shot is missing.
- The method has three training stages. What does it mean by end-to-end learning as described in several places in the paper? It seems each components are trained separately.
- The approach to get identity-agnostic feature is only to augment the images with color jitter, scaling and affine transformation. Such augmentations do not remove the identity information but only change the basic appearance of the facial images. The reviewer also believes such simple augmentations are not innovative.
- The app encode in Fig 2 seems to be not described.
- While the method shows plausible performance on the tested datasets, the generated video seem not to be tested with widely-used metrics like FVD.

**Questions:**

Please see the weakness.

**Details Of Ethics Concerns:**

The portrait animation has potential ethical impact. The authors briefly discuss it in P10.

---

> ### Author Response · Authors · 2024-11-19
> **Response to Reviewer qbkt**
>
> Thank you for your constructive comments. We have provided detailed responses to your questions as follows.
>
> **Re: Definition of Zero-Shot**
>
> In our work, zero-shot refers to the model’s capability to generalize to unseen portrait identities and driving motions without requiring fine-tuning during inference. Specifically, although our model is trained exclusively on real human videos in a self-driven manner, it can effectively generalize to out-of-distribution reference images, such as stylized or non-human characters (e.g., 3D cartoon portraits), and in-the-wild facial motions, for cross-identity driving. This definition of zero-shot aligns with those employed in many prior works, including X-Portrait (SIGGRAPH 2024), GAIA (ICLR 2024), DiffPortrait3D (CVPR 2024), and Zero-1-to-3 (ICCV 2023).
>
> **Re: Definition of End-to-End Training**
>
> X-NeMo represents an end-to-end fully differentiable and trainable pipeline from video to video, without relying on  predefined intermediate motion descriptors to condition the generator as in previous methods. While our full pipeline is trained in three stages for training efficiency and warm-up pretraining, it supports fully end-to-end optimization with all modules trainable.
>
> Specifically, in the first stage of image pretraining, we optimize both the backbone UNet and the reference net, which specializes the generative network for portrait generation. In the second stage, we simultaneously optimize all network components except for the temporal modules. Here, we achieve what we refer to as end-to-end learning of the motion encoder and image generator. The temporal module is trained independently in the third stage, solely to ensure cross-frame coherence, following the standard practices of prior works (e.g. AnimateDiff, AnimateAnyone, MagicAnimate).
>
> Our concept of end-to-end learning emphasizes optimizing the implicit motion representation directly within the generation process.   This  paradigm allows the motion encoder to fully exploit the motion diversity and richness inherent in the training video collections, without depending on external motion detectors. As a result, the learned motion embedding space captures fine-grained motion semantics effectively.
>
> **Re: Effectiveness and Novelty of Augmentation for ID Disentanglement**
>
> Thank you for your comments. We agree that applying color and spatial augmentations alone does not entirely remove the identity information encoded in the driving image. However, these augmentations effectively encourage the generative backbone to rely on the reference network for appearance and identity-related information.
>
> Moreover, our motion representation is designed as a 1-D, low-dimensional latent vector, which inherently restricts the information flow of high-frequency appearance and identity features from the augmented driving images. This compact representation focuses on capturing motion information, leading to reduced training errors and better disentanglement. This has been evidenced in our quantitative (Tab 1, 2) and qualitative results (Fig 1, 3 and video demo).
>
> While we acknowledge that data augmentation is a classical strategy, our approach uniquely leverages spatial augmentations within a diffusion-based framework for portrait reenactment. This is a key advantage of our method. In contrast, prior approaches relying heavily on spatially aligned control signals, such as with ControlNet or PoseGuider, are more sensitive to spatial augmentations, which can degrade both robustness and accuracy.
>
> **Re: Definition of App-Encoder in Fig.2**
>
> Thank you for pointing this out. As mentioned in line 356, the App-Encoder is a feature extractor network that encodes the reference image into an appearance latent embedding, $f_{\text{app}}$, which is utilized by the GAN branch to generate images. We will update the caption of Fig. 2 to include this description for improved clarity.
>
> **Re: FVD metric**
>
> Thank you for your suggestion. We have added comparisons of the FVD metric under the cross-reenactment setting with other baseline methods:
>
> |     | PD-FGC | LivePortrait | X-Portrait | FYE   | AniPortrait | Ours  |
> |-----|:--------:|:--------------:|:------------:|:-------:|:-------------:|:-------:|
> | FVD$\downarrow$ | 441.8  | 279.2        | 237.3      | 360.9 | 290.5       | 152.8 |
>
> The results demonstrate that our method achieves significantly lower FVD values, indicating a high level of temporal consistency and visual fidelity in the generated videos. This also aligns with the qualitative comparisons shown in the supplementary video.

---

> > ### Comment · Reviewer_qbkt · 2024-11-23
> >
> > The reviewer feels some concerns are addressed by the response, particularly the Definition of Zero-Shot, Definition of App-Encoder in Fig.2 and FVD results. The reviewer appreciates the detailed explanation.
> >
> > However, the reviewer is still concerned with the term end2end and the augmentation part. The description of ``end-to-end'' is not typical and may be misleading, with modules trained separately and finetune adaptively.  The authors are encouraged to reconsider their claim.
> >
> > For the augmentation part, the reviewer believes Augmentation does NOT achieve ID Disentanglement. Although augmentation is effective, as shown by many previous works even at the age of GANs, the color and spatial augmentations cannot isolate identity of the reacted human. The claim "these augmentations effectively encourage the generative backbone to rely on the reference network for appearance and identity-related information" is in dispute. Those identity-agnostic information like pose and facial expression is remained in the augmented images. The reviewer may argue that the performance boost is attributed to augmented distribution of the video dataset, which is high-dimension, complex and with relatively limited samples. This is similar to the traditional detection, image translation or domain adaptation tasks where the training distribution is complex but having a small number of samples benefits significantly from augmentation. And the proposed augmentation strategy may contribute nothing new to the community.
> >
> > The author believe a 1-D, low-dimensional latent vector inherently restricts the information flow of high-frequency appearance and identity features. This claim is not supported by experiments on other latent vector setting. The design of cross-attention to substitute the spatial addition operation is not new. Which operation is better is in dispute in the community like DiT exemplifies that the spatial addition with affine is better. The design mechanism does not provide new insight for the problem.
> >
> > ControlNet or PoseGuider, which are claimed by the authors to be more sensitive to spatial augmentations, are able to be integrated with augmentation simply by applying the same spatial augmentations to the condition and the target image. Moreover, the inconsistency between the condition and the image can be handled by the proposed cross-attn operation and other well-design pipelines for controllable generation, based on existing works like MIGC.
> >
> > Based on the above discussion, the reviewer thinks the main concerns are not fully addressed, and the manuscript in the current version is not suitable for publication. The reviewer choose not to change the rating.

---

> ### Author Response · Authors · 2024-11-24
> **Response to Reviewer qbkt (Part 1/2)**
>
> We appreciate your detailed feedback and would like to further address your concerns:
>
> **Re: Definition of End-to-End Training**
>
> Thank you for your comment. We will re-clarify the definition of "end-to-end" in the final version of the paper to avoid potential misunderstandings.
>
> Here, our use of "end-to-end training" emphasizes two key aspects:
>
> 1. Direct optimization of the implicit motion representation within the generation process, from video to video, distinguishing our approach from prior methods that rely on external motion detectors. That being said, our method enables directly learning new facial motion distributions from RGB face videos, without reliance on explicit motion representations.
>
> 2. After the initial three-stage training process (intended as warm-up), we train the entire pipeline end-to-end, with all trainable modules optimized simultaneously.
>
> We will ensure our revised description is more precise and reflects these points clearly in the final manuscript.
>
>
> **Re: Effectiveness and Novelty of Augmentation for Disentanglement**
>
> Thank you for your thoughtful feedback. We appreciate the opportunity to clarify our augmentation strategy and its role in our framework.
>
> Regarding your concern that *“color and spatial augmentations cannot isolate the identity of the reacted human. The claim ‘...' is in dispute. Those identity-agnostic information like pose and facial expression is remained in the augmented images”*, we suspect there may have been some misunderstanding about how augmentations are applied in our method. As illustrated in Fig. 2 of the paper, **augmentations are applied exclusively to the driving images, not the reference images**. The purpose of these augmentations is to disrupt identity-related information in the driving images (e.g., skin color, facial shape, and feature distribution) while preserving identity-agnostic information such as pose and facial expression, which are subsequently extracted by the motion encoder. During training, the inconsistency between the augmented driving image and the target image encourages the generative network to rely more heavily on the reference network to extract appearance- and identity-related information from the reference image.
>
> We acknowledge that augmentation alone is insufficient to achieve full disentanglement of identity and motion. As detailed in Sec. 3.3, augmentation is only one component of our multi-faceted strategy for disentanglement. Other key components include **1-D low-pass latent motion embedding,  Dual Head Latent Supervision and Reference Feature Masking**, which are critical in enabling the motion encoder to extract fine-grained, identity-agnostic motion features.
>
> We do **not** position the augmentation strategy as a technical contribution in the paper. While augmentation is conceptually similar to those used in GAN-based portrait animation approaches for distanglement, our framework uniquely combines it with **latent motion representations** under a **diffusion-based generative model**, yielding significant improvements in disentanglement. The core insight behind our approach is that **“generation facilitates understanding”**. By leveraging the superior generative capacity of diffusion models, our method enhances the motion encoder’s ability to embed the disentangled representation of facial images. This is evident in our comparative experiments (Fig. 5 and Tab. 1), where the GAN-based **PD-FGC** model trained on the same dataset shows inferior performance in both identity disentanglement and expression fidelity compared to our method.
>
> Nevertheless, we acknowledge that our combined strategies **do not guarantee 100% disentanglement** between identity and motion. However, our approach demonstrates **state-of-the-art disentanglement** for portrait animation, as evidenced by diverse qualitative evaluations and quantitative metrics, substantially outperforming prior works in this area.

---

> > ### Author Response · Authors · 2024-11-24
> > **Response to Reviewer qbkt (Part 2/2)**
> >
> > **Re: Spatial augmentations for ControlNet**
> >
> > Regarding your suggestion that "ControlNet or PoseGuider... are able to be integrated with augmentation simply by applying the same spatial augmentations to the condition and the target image," we respectfully disagree. For the portrait animation task, applying identical spatial augmentations to both the condition (driving) and target images would introduce discrepancies in facial structure between the reference image and the augmented target image. This discrepancy would lead to the model learning to generate identity attributes that differ from the reference image, which conflicts with the primary goal of portrait animation — maintaining **identity consistency** with reference image while transferring motion from driving images.
> >
> > On the other hand, if augmentations are applied exclusively to driving images, as done in our method, but the motion is integrated through ControlNet or PoseGuider, it will then inherently compromise motion accuracy and expressiveness for identity disentanglement. This has been extensively ablated in Fig. 3(a) and Tab. 2.
> >
> > **Re: Novelty of Cross-attention**
> >
> > We acknowledge that the cross-attention mechanism itself is not novel within the broader context of conditional diffusion models. However, its application in our work offers a unique and effective solution to key challenges specific to **video-based portrait animation**. Our method is the first to integrate latent motion representations with a diffusion model via cross-attention, marking a departure from the traditional ControlNet-based motion control paradigm. Through analytical reasoning (Sec 3.2) and extensive quantative (Tab. 2) and qualitative experiments (Fig.3 (a)), we demonstrate that cross-attention effectively mitigates identity leakage, a persistent challenge in **cross-identity portrait animation**, compared to spatial addition mechanisms.

---

> ### Author Response · Authors · 2024-11-25
> **Have our responses addressed your concerns？**
>
> Dear Reviewer qbkt:
>
> Thanks for your valuable comments. We have carefully considered your concerns and questions, and we have made every effort to address them thoroughly in our rebuttal. Since the discussion phase is halfway around, we would like to know if our responses have adequately addressed your concerns.
>
> We kindly ask the reviewer to reassess the paper in light of our responses, if they clear things up. If not, we would be happy to provide alternative explanations or clarifications on any part of the paper.
>
> Once again, we thank you for your time and effort in reviewing our paper and appreciate your feedback.

---

> > ### Comment · Reviewer_qbkt · 2024-11-30
> >
> > The reviewer is sorry for the late reply. The reviewer reads the detailed response from the authors and also skim the discussion between the authors and other reviewers. The reviewer's concerns are not addressed at this stage and chooses to remain the score unchanged. Here are the reasons.
> >
> > The reviewer's concern on color and spatial augmentations is not addressed. The reviewer is grateful for that the authors clarify the confusion on where the augmentation is applied, and the reviewer is sorry for the misunderstanding.
> >
> > However, the argument that augmentations "disrupt identity-related information in the driving images (e.g., skin color, facial shape, and feature distribution) while preserving identity-agnostic information such as pose and facial expression" is not convincing. Other approaches like pose estimation and face expression extraction are more likely to preserving identity-agnostic information. Beside, it is counter-intuitive that color augmentations disrupt skin color or spatial augmentations disrupt facial shape. The reviewer acknowledges that augmentation extends feature distribution and the inconsistency causes the network to extract more information, but this is not disentanglement yet a self-supervision strategy.
> >
> > Although the authors do not position the augmentation strategy as a technical contribution in the paper, Table 2 shows augmentation is one main cause of the performance improvement.
> >
> > The authors believe this is a unique combination to improve disentanglement. The reviewer fails to find estimation of disentanglement in the current version, and believe adopting a widely-used trick in the specific task is not new.
> >
> > The authors highlight the “generation facilitates understanding” insight, but such insight is widely reflect by papers with "analysis-by-synthesis".
> >
> > The authors acknowledge that the cross-attention is not novel. However, in Table 2 this is also the largest source of performance gain. This may show the performance gain is not attributed to novel designs.
> >
> > While the authors emphasize the performance gain, the reviewer is concerned about the novelty as echoing existing tricks in new tasks is not a good way to advance the research community. Therefore, the reviewer choose not to change the score.

---

> > > ### Author Response · Authors · 2024-12-03
> > > **Response to Reviewer qbkt (Part 1/2)**
> > >
> > > Thank you for your feedback. We are pleased that in our previous discussions, through revising the manuscript, adding more numerical experiments, and providing more detailed explanations, we were able to address your concerns regarding the definition of "zero-shot," the definition of "end-to-end training," the definition of the App-Encoder in Fig.2, and the FVD results. Based on our understanding, the remaining points of confusion seem to focus primarily on the augmentation strategy, which can be broken down into two main concerns:
> > > 1. You believe that the disentanglement demonstrated by our method relies primarily on augmentation, which limits the novelty of our approach.
> > > 2. You are concerned that the augmentation strategy is insufficient to fully remove identity-related information from the driving image.
> > >
> > > Below we provide a more detailed explanation to address these concerns according to your last comment.
> > >
> > >
> > > **Re: Evaluation of Disentanglement**
> > >
> > > > The reviewer fails to find estimation of disentanglement in the current version
> > >
> > > For the Portrait Animation task, **researchers [1, 2, 3, 4] commonly use cross-identity animation to demonstrate the disentanglement of motion and appearance.** Accordingly, in our paper and video, we provide extensive quantitative and qualitative results of cross-Reenactment to demonstrate that **our method achieves SOTA disentanglement**. These cross-Reenactment results show that the generated images maintain good motion consistency with the driving image and do not exhibit identity leakage. Therefore, we believe that our trained motion encoder effectively removes identity-related information from the driving image while simultaneously preserving identity-agnostic information.
> > >
> > > **Re: Role of Augmentation**
> > >
> > > > Although the authors do not position the augmentation strategy as a technical contribution in the paper, Table 2 shows augmentation is one main cause of the performance improvement.
> > >
> > > **The effectiveness of the disentanglement is demonstrated by ensuring both motion consistency between the generated results and the driving image, and identity consistency with the reference image.** While Tab.2 and Fig.3 show that the augmentation strategy plays an important role in preserving identity consistency, **ensuring identity consistency alone is NOT enough for the portrait animation task** (simply cloning the reference image can also maintain perfect identity consistency). Achieving fine-grained motion transfer is equally important. As shown in Fig.4 and the EMO-SIM metrics in Tab.2, our proposed Dual Head Branch strategy plays a crucial role. By comparing the EMO-SIM metrics in Tab.1 and Tab.2, we can see that the Joint Training and Reference Feature Masking strategies further ensure that our method significantly outperforms others in terms of motion expressiveness.
> > >
> > > In summary, the Dual Head Branch, Cross-Attention, and Augmentations are essential for achieving effective disentanglement. Furthermore, Joint Training and RFM enhance motion extraction, capturing subtle micro-expressions. **Therefore, disentanglement does not rely solely on augmentation.**
> > >
> > > **Re: Effectiveness of Augmentation**
> > > >The reviewer acknowledges that augmentation extends feature distribution and the inconsistency causes the network to extract more information, but this is not disentanglement yet a self-supervision strategy.
> > >
> > > **We agree that augmentation is a self-supervision strategy and acknowledge that it alone cannot achieve full disentanglement.** We are glad that you agree that the inconsistency between the target image and the augmented driving image leads to the motion encoder extracting more motion-related information during training.
> > >
> > > > The authors believe this is a unique combination to improve disentanglement. The reviewer believes adopting a widely-used trick in the specific task is not new.
> > >
> > > Although augmentation is a widely used trick, we would like to point out that for diffusion-based portrait animation methods, using spatial augmentation to promote disentanglement is **not straightforward**. Existing explicit-motion-based portrait animation methods have shown that **applying spatial augmentation alone does not effectively address the identity leakage issue (as seen in X-Portrait[1])**. Through the numerical results in Tab.2 and the visualizations in Fig.3, we aim to demonstrate that, while current methods have not fully explored the effectiveness of Spatial Augmentation, our proposed joint framework—combining a latent motion representation with a diffusion model—**shows the capability for enhancing reenactment performance by properly incorporating augmentations into a diffusion backbone.**

---

> > > ### Author Response · Authors · 2024-12-03
> > > **Response to Reviewer qbkt (Part 2/2)**
> > >
> > > **"Analysis-by-Synthesis" and our framework novelty**
> > > > The authors highlight the “generation facilitates understanding” insight, but such insight is widely reflect by papers with "analysis-by-synthesis".
> > >
> > > As we discussed in the Related Work section, at present, due to the stronger generative capabilities of diffusion models, recent portrait animation methods focused on designing diffusion-based frameworks to improve performance across various portrait styles. However, **integrating the "analysis-by-synthesis" approach into the diffusion model is not trivial**. During the training process, the diffusion-based generator cannot directly produce fully denoised images, and as a result, it **cannot directly leverage image-level losses** (such as identity- or expression-consistent losses, or contrastive losses) as previous methods have done to optimize the encoder and assist disentanglement.
> > >
> > > Although the "analysis-by-synthesis" approach is a classic concept (i.e., learning latent representations by fitting the dataset), to our knowledge, no prior diffusion-based methods have proposed using this idea to improve reenactment performance, nor have they offered solutions to the above disentanglement problem. Instead, they have had to abandon "analysis-by-synthesis" and rely on offline detectors to extract immediate motion representations, which suffer from limited expression capability and structure-related identity leakage.
> > >
> > > In this paper, by proposing the novel Dual Head Branch framework, we introduce additional image-level losses and successfully **combine latent motion representation with a diffusion model for the first time**. Through ablation experiments and numerous cross-driving demos, we demonstrate that our self-supervised learning strategies—consisting of "Augmentations for driving image + Cross-Attention-based Motion Injection + encoder-generator Joint Optimization + Reference Feature Masking"—effectively achieves disentanglement and significantly enhances the motion encoder’s ability to capture expressions.
> > >
> > > By realizing the "analysis-by-synthesis" approach, we have proposed **a scalable self-supervised learning framework based on the diffusion model**, which means that in the future, we can pre-train on larger portrait video datasets, laying the foundation for broader applications in portrait video generation. We hope that the reviewers will evaluate the overall novelty of our work from the perspective of global framework design.
> > >
> > > **Other Mentioned Problems**
> > > >it is counter-intuitive that color augmentations disrupt skin color or spatial augmentations disrupt facial shape
> > >
> > > During training, we augment the driving image with color augmentation (color jitter) and spatial augmentation (scaling and affine transformation). The former partially alters appearance-related information (e.g., skin color), while the latter partially changes structure-related information (e.g., facial shape). This is common practice in portrait animation. You can view the visualized augmented images in our Fig.2 or in the pipeline figure from previous works, such as [4].
> > >
> > > >The authors acknowledge that the cross-attention is not novel. However, in Table 2 this is also the largest source of performance gain. This may show the performance gain is not attributed to novel designs.
> > >
> > > For diffusion-based portrait animation methods, introducing cross-attention-based motion injection is not straightforward. Existing explicit-motion-based portrait animation methods that directly use cross-attention for motion injection do not show significant performance improvements (e.g., AnifaceDiff[5]). It is precisely by combining this with the latent motion representation used in our method that we are able to effectively mitigate the identity leakage issue.
> > >
> > >
> > > ---
> > > [1] Xie, You, et al. "X-portrait: Expressive portrait animation with hierarchical motion attention." SIGGRAPH 2024.
> > >
> > > [2] Drobyshev, Nikita, et al. "EMOPortraits: Emotion-enhanced Multimodal One-shot Head Avatars." CVPR 2024.
> > >
> > > [3] Drobyshev, Nikita, et al. "Megaportraits: One-shot megapixel neural head avatars." ACMM 2022.
> > >
> > > [4] Burkov, Egor, et al. "Neural head reenactment with latent pose descriptors." CVPR 2020.
> > >
> > > [5] Chen, Ken, et al. "AniFaceDiff: High-Fidelity Face Reenactment via Facial Parametric Conditioned Diffusion Models." arXiv:2406.13272.

---

### Official Review · Reviewer_82UL · 2024-11-03

**Soundness:** 3
**Presentation:** 3
**Contribution:** 3
**Rating:** 8
**Confidence:** 5

**Summary:**

This work presents a portrait animation method that diverges from traditional landmark-based approaches. It leverages a latent motion descriptor enhanced by a low-pass filter and incorporates motion priors through cross-attention, eliminating the reliance on aligned pose information. To develop a robust and fine-grained motion descriptor, the method includes a GAN head and employs techniques such as data augmentation and masked modeling.

**Strengths:**

1. This work proposes a feasible solution to address the limitations of previous portrait animation methods that rely on explicit motion descriptors or the integration of motion information through PoseGuider and ControlNet.
2. This study demonstrates strong visual performance across various samples, showcasing its robust capabilities in motion transfer and stability.
3. This work includes comprehensive comparisons with prior methods and an ablation study to validate the proposed techniques.

**Weaknesses:**

1. A temporal evaluation of spatially aligned motion injection versus attention-based motion injection is recommended. Intuitively, spatially aligned motion injection is expected to provide better temporal consistency due to its stronger spatial priors.
2. Additional analysis and experiments are needed to clarify why X-NeMo achieves such high levels of temporal consistency. Other methods, such as LivePortrait and X-Portrait, also include a stage for training temporal modules, yet they still exhibit some flickering in their results. The paper mentions only temporal modules and prompt traveling as means to achieve temporal consistency, but I remain unclear about the source of X-NeMo's superior performance in this regard.

If these concerns are adequately addressed, I will consider raising my score to "accept."

**Questions:**

1. In the quantitative comparisons, do the results for AniPortrait and X-Portrait come from the officially released weights or from weights that were retrained on your training dataset? If the latter, why do the results for both methods perform poorly in cases involving tongue motion, considering that the NerSemble dataset should include training samples with tongue motion? This is particularly concerning for X-Portrait, which is also a non-landmark method.

---

> ### Author Response · Authors · 2024-11-19
> **Response to Reviewer 82UL (Part 1/2)**
>
> Thank you for your thoughtful feedback and for recognizing the visual quality and comprehensiveness of our experiments. We provide our responses to your comments as follows.
>
> **Re: Temporal Evaluation: Spatial-Aligned  v.s. Attention-Based Motion Injection**
>
> Thank you for raising this insightful question.
> In our submission, we have conducted an ablation study (referenced as “w/o cross-attn” in Table 2), where the motion latent is transformed into a 2D control map via an upsampling decoder and applied to the UNet through ControlNet. To further substantiate our claims, we include FVD metric for evaluating video fidelity and temporal consistency. The results are as follows:
>
> | Method | ID-SIM$\uparrow$ | AED/APD$\downarrow$ | EMO-SIM$\uparrow$ | FVD$\downarrow$ |
> |------------|----------|---------|---------|-------|
> | Ours | 0.787 | 0.039/3.42 | 0.65 | 152.8 |
> | w/o cross-attn | 0.697 | 0.040/3.55 | 0.48 | 191.7 |
>
> As these results indicate, our attention-based motion injection mechanism achieves superior FVD scores, highlighting better temporal coherence and visual fidelity.
>
> Using spatially aligned motion injection, however, leads to severe **ID leakage issues**, as illustrated in Fig. 3(a) of the paper. As analyzed in the paper, spatially aligned motion injection inherently learns to formulate the image-wise structural guidance, including identity-specific characteristics, before the denoising backbone. This violates our design principle that the motion control path should remain agnostic to identity-specific attributes. Such inconsistency is particularly noticeable when driving with continuous exaggerated expressions or large head rotations, further degrading temporal consistency.
>
> Moreover, the baseline with spatially aligned motion injection **does not inherently provide stronger spatial priors**, as the 2D control map is still derived from the 1D motion latent. In contrast, our attention-based motion injection mechanism effectively mitigates the identity leakage and leverages fine-grained latent motion embeddings to generate smooth and coherent animation results.
>
> **Re: Superior Temporal Consistency Achieved by X-NeMo (vs. AniPortrait and X-Portrait)**
>
> Thank you for your valuable feedback and for acknowledging the temporal consistency achieved by our method. While LivePortrait is a GAN-based method, we focus here on the differences between our approach and AniPortrait and X-Portrait.
>
> To ensure temporal consistency, our method incorporates strategies similar to those used in prior works, such as the training of temporal modules, the prompt traveling strategy during inference, and the use of an SVD VAE decoder for multi-frame decoding. However, X-NeMo achieves superior temporal consistency primarily due to the expressive capabilities of our motion latent descriptor, which is encoded directly from the original driving frames.
>
> Our end-to-end training framework allows the motion encoder to extract fine-grained and temporally smooth motion control signals. In contrast, AniPortrait and X-Portrait rely on pretrained motion control signals, such as explicit landmarks (AniPortrait) or implicit landmarks (X-Portrait). These signals are prone to instability and jitter across frames, and are not sufficient to fully capture the complex facial expressions present in the training dataset (e.g., NerSemble). Consequently, the conditional diffusion generators in these methods tend to retain higher levels of noise-induced variability, leading to instability during the reenactment process.

---

> > ### Author Response · Authors · 2024-11-19
> > **Response to Reviewer 82UL (Part 2/2)**
> >
> > **Re: Tongue motion by AniPortrait and X-Portrait**
> >
> > For AniPortrait, we retrained the model on the same dataset as ours, while for X-Portrait, we utilized the officially released pre-trained model (as the training code is not available). Both methods inherently struggle to generate robust tongue motion due to limitations in their motion control signals, which fail to capture the intricacies of tongue movements.
> >
> > Although the NerSemble dataset includes cases with tongue motion, the sparse landmarks employed by AniPortrait do not account for tongue dynamics. Consequently, during training, the motion control signals related to tongue movements are absent, leading to a coupling effect between tongue motion and other facial expressions. This coupling prevents AniPortrait from producing stable and robust tongue animations.
> >
> > For X-Portrait, the training data is synthesized using the off-the-shelf portrait animator Face-vid2vid, which is limited in handling complex expressions, including tongue movements. While X-Portrait operates without explicit landmarks, its training set does not include driving images with tongue motion, resulting in suboptimal performance on test cases involving such movements.
> >
> > In contrast, our method leverages an end-to-end learning framework for both motion encoding and visual generation. This approach allows our model to learn intricate expressions, such as tongue dynamics, directly from the training distribution, resulting in significantly improved performance on these challenging cases.

---

> ### Author Response · Authors · 2024-11-25
> **Have our responses addressed your concerns？**
>
> Dear Reviewer 82UL:
>
> We appreciate your valuable comments on our paper. We have carefully considered your feedback and provided detailed response to your concerns regarding temporal consistency and tongue motion. We hope that you find our rebuttal satisfactory.
>
> As the discussion phase is still in progress, we would like to invite you to review our rebuttal and share any further comments or questions that you may have. We value your feedback and are committed to improving the quality of our paper accordingly.
>
> If our rebuttal has successfully addressed your concerns, we kindly request you consider updating your score. Your updated evaluation would be greatly appreciated.
>
> Thank you for your time and attention.

---

> ### Comment · Reviewer_82UL · 2024-11-25
>
> Thanks for the explanation which partially address my concerns. However, the exact contribution source for the performance is still in question.  First, I do not think FVD is good metric to evaluate the temporal coherence as it is a video-level metric to measure the overall video quality. What I concerns are ① the periodic "flicking" artifact (only occurs in the overlapped frames between two clips) introduced by the clip-based inference strategy adopted almost by all existing video generation methods, and ② the "stitching" artifact (i.e., not smooth and consistent enough across frames and even within an image) almost appearing in all other methods except for X-Nemo.
>
> One possible reason of such good performance may come from the strong single-frame face reenactment performance, which can thus lead to a strong video-level performance. However, I am not agree with what the authors' claim on the contribution source of the performance. The authors mention "X-NeMo achieves superior temporal consistency primarily due to the expressive capabilities of our motion latent descriptor", and thank to their "end-to-end training framework allows the motion encoder to extract fine-grained and temporally smooth motion control signals".  From the method and experimental analysis, I guess the contribution sources are the GAN head, augmentation and stepwise training strategy.
>
> First, the verification of the effectiveness of the GAN head and augmentation can be drawn to some extent in Table 2. The GAN head and augmentation are the most significant factor influencing AED and APD, which reflects the motion accuracy including the expression distance and head pose distance.
>
> Second, the authors' claim of the end-to-end training is in question. X-Nemo is trained in several stages, in which the first stage trains the U-Net and reference net while the second stage trains the motion encoder and motion-attention layers. I think the separation of appearance encoding and motion encoding (previous works almost learn the two in one step), as well as the unification of motion encoding and motion injection (also named as "end-to-end training" by the authors), together with augmentation and the GAN head providing another supervision to force the model to entangle the appearance and motion such clearly.
>
> I acknowledge the contribution of this paper, but deep analysis are still required to refine this paper, especially more clear explanation about the contribution source of the strong performance. Current writing will mislead others in understanding which are the important techniques in training a good portrait animation model. I suggest to accept this paper. Of course, better version is required. If the authors promise to provide a revised version according to the discussions, I will raise my score.

---

> > ### Author Response · Authors · 2024-11-26
> >
> > Thank you for your valuable feedback and for recommending our paper for acceptance. We sincerely appreciate your constructive suggestions and will revise the manuscript to clarify the sources contributing to X-NeMo's strong performance.
> >
> > We acknowledge that FVD is not a perfect metric for evaluating temporal coherence, and we are unaware of a universally accepted alternative. However, if a more suitable metric becomes available, we would be happy to include it in the final manuscript.
> >
> > Regarding the absence of "periodic flickering" artifacts in X-NeMo,  we have observed that using **a prompt traveling strategy effectively mitigates periodic flickering artifacts**, as can be seen in the supplemental video. Neither our method nor X-Portrait exhibits noticeable periodic flickering.
> >
> > As for the absence of **stitching artifacts**, we believe this aligns with the reasoning summarized in our initial response, which also echoes your observation: “Strong single-frame face reenactment performance leads to strong video-level performance.”
> >
> > We fully agree that end-to-end training alone is **not** the primary contributor to the strong performance of single-frame face reenactment. In our initial response, the statement “*Our end-to-end training framework allows…*” was intended to highlight a key difference in framework design between our method and prior approaches like AniPortrait and X-Portrait. While these methods rely on **pre-trained motion detectors**, we adopt a **learnable motion representation** that is optimized jointly with the diffusion generator during training. This fundamental design inherently enables our model to achieve better expressive performance than previous methods, provided motion and identity are disentangled more effectively. However, **we did not mean to suggest that end-to-end learning alone is the most critical factor for disentanglement**. Instead, as described in Line 309 of the paper, “*straightforward self-supervised training of the entire framework does not inherently disentangle identity from facial motion*”. Actually, besides end-to-end training , several complementary components, as outlined in Sec. 3.2 and Sec. 3.3, jointly contribute to disentanglement, including: Cross-Attention-Based Motion Injection, GAN Head Supervision, Augmentations, Reference Feature Masking (RFM).
> >
> > As shown in Table 2, **GAN Head Supervision, Cross-Attention, and Augmentations** play essential roles in achieving reasonable motion and identity disentanglement. Meanwhile, **end-to-end training and RFM** further refine motion extraction, allowing the model to capture subtle micro-expressions (not captured by AKD but reflected in the EMO-SIM metric) and enabling vivid and expressive reeactment results.
> > Regarding our stepwise training strategy: in the second stage of training, we do not only train the motion encoder and motion-attention layers. Instead, **we simultaneously optimize all network components except for the temporal modules**. In our practice, we did not observe any significant improvement resulting from separating appearance encoding and motion encoding during training. In fact, after completing the three-stage warm-up process, the entire pipeline is fine-tuned end-to-end, with all trainable modules optimized simultaneously. We will re-clarify the training stages in the final version of the paper to prevent potential misunderstandings. Additionally, we can include an ablation study to further explore the effects of the stepwise training strategy.
> >
> > To better address your insightful feedback and help future researchers understand the critical components of X-NeMo’s success, in the final revision we will:
> > 1. Provide a detailed explanation of each component's contribution to disentanglement and temporal consistency.
> > 2. Re-clarify the training stages for improved transparency.
> > 3. Include additional ablation studies to analyze the impact of the stepwise training process.
> >
> > Thank you again for your valuable comments. We are confident that these revisions will enhance the clarity and robustness of our paper.

---

> ### Comment · Reviewer_82UL · 2024-11-26
>
> A qualitative comparison with clear clarification—such as selecting frames from the overlapping clip, zooming in, or employing other effective presentation techniques—could effectively demonstrate the superiority of X-NeMo in terms of temporal consistency if there is no suitable metrics for assessing temporal consistency. Of course, It is important to avoid terms like "end-to-end training", as they have established meanings that are widely recognized.
>
> I look forward to your updated version with the promised refinements, as I believe it will provide valuable insights to the community. I am willing to increase my score to accept the strong performance and design of the method.

---

> ### Author Response · Authors · 2024-11-27
>
> Thanks for your recognition of the contributions of our work. We are pleased to hear that, after our previous discussion, it seems that your concerns have been well addressed. We sincerely hope that you will raise your score to support our work. We are currently refining our work based on feedback from the last round of discussion:
> 1. **We have updated the paper with revisions indicated with blue text**:
>   - Revised several statements about the 'end-to-end training' in the original manuscript.
>   - Re-clarified the training stages.
>   - Added a section in the Appendix to explicitly clarify the contribution of each component to the overall model performance.
> 2. **We are conducting an ablation study** on the stepwise training process, which means we need to retrain a variant that simultaneously optimizes all parameters from scratch. However, due to the relatively small batch size, training this variant from scratch is very time-consuming. This experiment is ongoing, but given the rebuttal time constraints, we apologize that we may not be able to present the results immediately within the discussion period. We promise that we will include this ablation study in the final version of the paper.

---

### Official Review · Reviewer_jJFM · 2024-11-03

**Soundness:** 3
**Presentation:** 3
**Contribution:** 3
**Rating:** 8
**Confidence:** 4

**Summary:**

This paper proposes a novel portrait animation framework that extracts identity-free motion through a specially designed module and injects the motion using cross-attention, while utilizing GAN to enhance the accuracy of motion capturing. Extensive experiments demonstrate the effectiveness of this approach.

**Strengths:**

1.This paper is well written, easy to follow.
2.This paper proposes a new portrait animation pipeline that effectively addresses the longstanding issues of identity entanglement and loss of motion expressiveness.
3.Extensive experiments demonstrate the effectiveness of this method.
4.Great work! The motivation and experimental results for each component are solid. The demo in the supplementary materials also looks very impressive; (if it isn’t cherry-picked)

**Weaknesses:**

1.Since the motion model is trained, could it struggle to adapt to out-of-distribution (OOD) motions, could you provide extreme or unusual facial expressions to demonstrate robustness?
2.In the results provided in the paper and the demo, the facial features of the driving and reference are quite similar. Could you provide more examples where facial features (such as eyes, mouth, nose, etc.) or face position or head pose are inconsistent?
3.As stated in W2, I also cannot tell if this paper truly addresses the issue of identity leakage, as it appears that most of the features in the driving and reference images are quite similar, could you provide more convincing prof or experiments?

**Questions:**

1.For non-human data, existing methods find it challenging to crop the face. If you were to apply these methods to such data, what kind of solution would you propose?
2.Could you simply train a model with added OOD data to provide results for non-human cases?

---

> ### Author Response · Authors · 2024-11-19
> **Response to Reviewer jJFM**
>
> Thank you for your constructive feedbacks and insightful comments. We are glad to see that you acknowledged the effectiveness of our method and appreciated our visual quality. We provide our detailed responses to your questions as follows.
>
> **Re: Effectiveness of Our Method under OOD Motions**
>
> Our motion encoder, trained on a diverse video set of rich and complex facial expressions, effectively captures structured spatial variations and correspondences. As demonstrated in Figures 4 and 5, as well as in our supplementary videos, our method exhibits strong generalization capabilities, handling novel facial expressions and head movements sourced from in-the-wild movie clips and expression videos.
>
> To further address your concern, we added a new figure in the paper and included additional examples (Figure 13, first two rows) of OOD facial motions—such as extreme mouth distortion, eye-rolling, and lip puckering. Despite these exaggerated expressions being outside our training distribution, our method successfully replicates and transfers fine-grained expression details across subjects.
>
> That said, as noted in the Limitations section (Figure 7), our current model may struggle with highly exaggerated expressions that are entirely absent from the training data. However, the scalable nature of our end-to-end framework offers a clear path for improvement by incorporating additional correlated training videos. This flexibility contrasts with many prior approaches that rely on explicit motion representations (e.g., facial landmarks or synthetic cross-identity images), where the motion extraction often fails already under extreme expressions.
>
> **Re: Cross-Identity Reenactment with Distinct Identity Features**
>
> We have demonstrated a range of qualitative examples where the source and driving characters differ significantly in facial shapes, proportions, styles, and head orientations. Notable cases include 3D/2D cartoon characters driven by real human motion (Figure 1) and examples from our supplementary video, showcasing pronounced differences in head poses (timestamps: 1:06, 1:37, 2:24) and facial features (timestamps: 2:30, 4:33, 5:45). Furthermore, Figure 5 illustrates that while prior baselines show substantial drift in identity characteristics, our method achieves the highest fidelity in preserving identity resemblance.
>
> To further support our claims, we added a new figure in the paper and included additional visual evidence (Figure 13, last two rows: "Panda" and "Thanos"), which highlight extreme identity attribute differences. Thanks to our identity-disentangled design, our method consistently delivers high-fidelity reenactments with minimal identity leakage from the driving video.
>
> **Re: Non-Human Reenactment**
>
> Thank you for your insightful question. Our method’s independence from pre-defined intermediate representations grants us the flexibility to train directly on non-human video datasets after pre-processing. We believe that our approach is well-suited for video datasets focused on foreground object motion. In such scenarios, general object detectors like YOLO or SAM could be employed to isolate and crop the foreground objects for processing.
>
> However, as we have not yet collected or processed such a non-human video dataset, we regret that we cannot provide empirical results to validate this approach at this time. That said, we see significant potential in extending our framework to accommodate a broader range of video categories, and we intend to explore this direction in future work.

---

> > ### Comment · Reviewer_jJFM · 2024-11-25
> >
> > Could you provide more extensive and diverse cases of cross-identity and non-human reenactment?

---

> > > ### Author Response · Authors · 2024-11-26
> > >
> > > We added **a new figure** in the paper and included additional visual evidence (**Figure 13, last six rows**), which highlights extreme identity attribute differences. **Please re-download the latest version PDF to check these visual results.**

---

> > > > ### Comment · Reviewer_jJFM · 2024-11-26
> > > >
> > > > Could you provide a video demo? Additionally, can each reference be driven by a different driving video? For example, in Fig. 13, there could be 8×8=64 video combinations. Would it be possible to share demos of these videos?

---

> > > > > ### Author Response · Authors · 2024-11-28
> > > > >
> > > > > Thank you for your feedback.
> > > > >
> > > > > In response, we have added a new video to the **Supplementary Material** titled *rebuttal_sup_video.mp4,* where we demonstrate eight cross-identity video demos corresponding to Fig. 13. From these demos, you can observe that for most non-human identities, our model successfully maintains identity consistency while also ensuring precise motion control. However, as we discuss in the *Limitation* section (Sec. 5) of the paper, for extreme out-of-domain portraits—such as the cartoon character in the final video—we do notice some artifacts, such as blurred eyes or inconsistent appearances. We believe that by incorporating more non-human training data in the future, we can further improve the model’s generalization to various identity types.
> > > > >
> > > > > Due to time constraints and limited computational resources (as we are also running additional experiments based on other reviewer feedback), we are unable to provide more demos beyond the eight shown at the moment. However, after the paper's acceptance, we plan to release the pretrained model and code for the community to conduct extensive testing.

---

> > > > > > ### Comment · Reviewer_jJFM · 2024-11-28
> > > > > >
> > > > > > As the author mentioned, while X-Nemo struggles to generalize on OOD data, its other components have demonstrated strong effectiveness on human data. A key concern is whether the robustness of Sam or YOLO on OOD data is sufficient to support the expansion of large-scale non-human datasets—referring specifically to portraits that differ entirely from human facial features, such as TED bear or the panda shown in Figure 13. Overall, I recommend accepting this paper.

---

### Meta-Review · Area_Chair_ZLew · 2024-12-22

**Metareview:**

The authors propose X-NeMo, a zero-shot portrait animation method based on diffusion models. This animates an image of a face using movements from a driving video of a different person. They specifically focus on preserving identity of the input image and capturing the full range of facial expressions.
During the initial round of reviews, this submission received ratings of 8, 8, 3, 6.
After discussion with the reviewers, the final consensus was 2 reviewers voting for weak accept, 1 borderline, and 1 reject. While the reviewers appreciated the quality of outputs, they raised issues with the quality of writing and lack of exploration of what really helps improve the output quality.
After considering all the strengths and weaknesses of the submission, the AC recommends weak acceptance, based on the impressive quality of the results. The authors are encouraged to improve the quality of the writing as well as ablations/insights into the proposed method.

**Additional Comments On Reviewer Discussion:**

Rebuttal:
Reviewer jJFM asked about the model's quality on OOD data.
Reviewer 82UL requested additional analysis and ablations on which changes introduced help achieve the quality improvements, and ultimately felt that the authors were not able to provide all the sufficient information during the time frame.
Reviewer qbkt raised a number of concerns regarding novelty and contributions, and the authors attempted to provide a sufficient defense of their work.

Discussion with AC:
Reviewer 82UL and jJFM appreciated the quality of outputs but remained unhappy with the writing and lack of ablations, ultimately advocating for weak accept. Reviewer tmTm decided to be borderline due to the lack of good writing. Reviewer qbkt advocated for rejection owing to the lack of perceived novelty and analysis.

---

### Decision · Program_Chairs · 2025-01-22

Accept (Poster)